# Dust radiative forcing in CMIP6 Earth System models: insights from the AerChemMIP piClim-2xdust experiment

Ove W. Haugvaldstad[1,2], Dirk Olivié[1], Trude Storelvmo[2], and Michael Schulz[1,2]

[1]The Norwegian Meteorological Institute, Oslo, Norway
[2]University of Oslo, Department of Geoscience, Oslo, Norway

**Correspondence:** Ove W. Haugvaldstad (oveh@met.no)

**Abstract.** Mineral dust significantly affects the downwelling and upwelling shortwave (SW) and longwave (LW) radiative fluxes, and changes in dust can therefore alter the Earth's energy balance. This study analyses the dust effective radiative forcing (DuERF) in nine CMIP6 Earth System Models (ESMs) using the *piClim-2xdust* experiment from AerChemMIP. The *piClim-2xdust* experiment uses a global dust emission tuning factor to double the emission flux. The DuERF is decomposed into contributions from dust-radiation (direct DuERF) and dust-cloud (cloud DuERF) interactions. The net direct DuERF ranges from $-0.56$ to $0.05\,\mathrm{Wm}^{-2}$. Models with lower (higher) dust absorption and a smaller (larger) fraction of coarse dust show the most negative (positive) direct DuERF. The cloud DuERF is positive in most models, ranging from $-0.02$ to $0.2\,\mathrm{W\,m}^{-2}$, however, they differ in their LW and SW flux contributions. Specifically, NorESM2-LM shows a positive LW cloud DuERF attributable to the effect of dust on cirrus clouds. The dust forcing efficiency varies tenfold among models, indicating that uncertainty in DuERF is likely underestimated in AerChemMIP. There is a consistent fast precipitation response associated with dust decreasing atmospheric radiative cooling (ARC). Models with strongly absorbing dust show reduced precipitation, explainable by decreased clear-sky ARC (up to $3.2\,\mathrm{mm\,year}^{-1}$). In NorESM2-LM, this decrease is associated with a cloudy sky ARC due to an increase in cirrus clouds (up to $5.6\,\mathrm{mm\,year}^{-1}$). Taken together, these findings suggest that the fast precipitation response induced by dust alone may be significant and comparable to that caused by anthropogenic black carbon.

## 1  Introduction

Mineral dust aerosols (hereafter referred to as 'dust') are highly abundant in the atmosphere and represent the dominant aerosol species in terms of mass loading (Kok et al., 2021). The most important dust sources are located in the Northern Hemisphere, specifically within the arid and semi-arid regions of Northern Africa, the Middle East, Central Asia, and East Asia (Kim et al., 2024). Dust emission is governed by surface winds, but is also influenced by environmental factors such as soil moisture, temperature, and precipitation (Zhao et al., 2022). Dust causes a diverse set of radiative effects that influence the energy balance of the top-of-the-atmosphere (TOA): it modulates radiation through scattering and absorption of longwave (LW) and shortwave (SW) radiation (e.g., Kok et al., 2017; Myhre and Stordal, 2001; Claquin et al., 1998), it indirectly influences cloud formation by acting as cloud condensation nuclei (CCN) or ice nucleating particles (INP) (e.g., Froyd et al., 2022; Koehler et al., 2009), and significantly alters the concentration of other atmospheric pollutants through heterogeneous chemistry (e.g., Soussé Villa

et al., 2025; Cwiertny et al., 2008; Bauer et al., 2007). Furthermore, dust alters surface reflectivity by changing the albedo of snow and ice surfaces upon deposition (e.g., Shi et al., 2021; Tuccella et al., 2021). The high complexity of the various dust radiative effects makes quantitive estimates of the TOA radiative impact of dust uncertain (Kok et al., 2023). In addition to altering the TOA energy balance, changes in dust also influence the energetics of the atmosphere, which in turn affects precipitation (Miller et al., 2004). This influence occurs initially through a rapid response mediated by changes in tropospheric temperatures that impact atmospheric stability and then a slower response in terms of changes in surface temperature and evaporation (Zhang et al., 2021). Finally, dust may also alter atmospheric circulation and therefore dust emissions themselves through feedback loops, as has been discussed for the African Monsoon region (Evans et al., 2020; Pausata et al., 2016). Consequently, variations in dust burden could have significant climatic implications.

Substantial evidence indicating that atmospheric dust burden has significantly increased in several regions around the globe since the beginning of the industrial era has been established by observations (Hooper and Marx, 2018; Marx et al., 2024; Mulitza et al., 2010), with a recent reconstruction of changes in dust loading from 1850 until 2000 showing an increase in dust by around $55 \pm 30\,\%$ (Kok et al., 2023). However, state-of-the-art Earth System Models (ESMs) fail to represent this increase and, more importantly, miss the potentially important radiative forcing of increased dust and its interactions with radiation, clouds, atmospheric chemistry, snow, and ice (Leung et al., 2025; Kok et al., 2023). Recently, dust emission datasets have become available that ESMs can use to account for the historical increase in dust and quantify the dust effective radiative forcing (DuERF) (Leung et al., 2025). However, to tell whether these estimates of DuERF would be reliable, we need to know whether the ESMs can be trusted to represent the wide scope of dust radiative effects. Consequently, it is necessary to document how current ESMs represent the physical properties of dust and dust-related processes and to consider how differences between models in the representation of dust and its interactions contribute to the uncertainty in DuERF and other possible dust climate responses. A recent 2023 assessment of the dust effective radiative effect (DuERE) arrived at a median value of $-0.2\,\mathrm{W\,m^{-2}}$ with a 90% confidence interval ranging from $-0.7$ to $0.4\,\mathrm{W\,m^{-2}}$ (Kok et al., 2023). Furthermore, in 6 out of the 9 DuEREs included in this assessment, confidence with respect to the assessed value ranged from low to very low, highlighting a significant knowledge gap in the ESMs.

The direct DuERE is the radiative effect that is most accurately represented within ESMs, and the sources of uncertainties are generally well understood (Kok et al., 2023). Besides dust lifetime and emission strength, which remain unobservable variables, the uncertainty in direct DuERE is mainly related to four key factors: the complex index of refraction (CRI) (e.g., Myhre and Stordal, 2001; Li et al., 2021), the particle size distribution (PSD) within the atmosphere (e.g., Kok et al., 2017), dust LW radiative effects and in particular LW scattering (e.g., Dufresne et al., 2002), and the shape of the dust particles (e.g., Ito et al., 2021). The CRI largely governs the dust SW absorption and is related to the mineralogical composition of the dust particles (Di Biagio et al., 2019). The composition of dust is highly source dependent; however, including source-dependent values of CRI requires additional tracers, which substantially increases computational expense. Therefore, ESMs to date have often resorted to using a single global value for the dust CRI based on an average dust composition (Castellanos et al., 2024). Moreover, the CRIs of dust used in ESMs in the early 2000s (e.g., OPAC, Hess et al., 1998) are still in use in some ESMs today (e.g., NorESM2, MIROC6) and have been shown to overestimate SW dust absorption (Adebiyi et al., 2023b; Di Biagio

et al., 2019). Furthermore, replacement of OPAC CRIs with more recent regionally resolved CRIs from Di Biagio et al. (2020) led to a tripling (from $-0.24$ to $-0.78\,\mathrm{W\,m^{-2}}$) of the SW dust direct radiative cooling (Wang et al., 2024). The switch to observationally consistent CRI of hematite also increased the SW dust cooling (Li et al., 2024). However, updates of dust optical properties have not been done consistently across ESMs, which has contributed to the apparent persistently large inter-model spread dust mass absorption coefficient (MAC) and the single scattering albedo (SSA) (Gliß et al., 2021; Huneeus et al., 2011). The PSD of dust is also an important cause of uncertainty in DuERE (Adebiyi and Kok, 2020; Kok et al., 2017). Early on, ESMs often assumed that dust aerosols with particle diameters larger than 10 $\mu$m were too large to have a significant climate impact due to their short lifetime (Adebiyi et al., 2023a) and were therefore often neglected. However, later observations have shown that coarse to super-coarse dust—sensu Adebiyi et al. (2023a) $> 10, \leq 62.5\,\mu$m—is transported in non-negligible quantities further than expected (e.g., Ryder et al., 2018; Adebiyi et al., 2023a). Kok et al. (2017) showed that including super-coarse particles up to 20$\mu$m reduced the TOA DuERE by 50% (from $-0.46$ to $-0.2\,\mathrm{W\,m^{-2}}$) due to the shift of the PSD to larger sizes, reducing SW extinction while increasing LW warming. The impact of LW warming could be even larger as most models currently do not include LW scattering (Adebiyi and Kok, 2020), which has been shown to increase LW DuERE by up to 50–60% (Dufresne et al., 2002). Lastly, ESMs typically assume that dust is a spherical particle. Although this assumption is appropriate for fine dust particles, it can be very inaccurate for coarse to super-coarse dust, causing an underestimation of the surface-to-volume ratio, which leads to an overestimate of dry deposition (Ginoux, 2003) and an underestimation of extinction efficiency (Ito et al., 2021). Ito et al. (2021) found that dust asphericity alone increased the SW TOA cooling by around 15% ($-0.32$ vs. $-0.28\,\mathrm{W\,m^{-2}}$ on a global scale), however, asphericity had limited impact on net TOA DuERE due to increased LW warming. Despite the mentioned complexities, the current representation of direct DuERE in ESMs holds up well compared to the way that ESMs represent dust-cloud interactions.

Currently, there is a lack of consistency in how ESMs represent dust indirect effects on clouds, with state-of-the-art models showing fundamentally different results. For example, some ESMs treat dust as externally mixed and hydrophobic and conse-quently, dust is not considered a CCN and thus does not have an indirect effect on warm clouds (e.g., CNRM-ESM2-1, Michou et al., 2020). Among models that consider dust to be a CCN, there are differences in dust CCN efficiency. For example, a common approach in ESMs is to consider freshly emitted dust to be insoluble, but to allow the dust to be transferred from an insoluble to a soluble state through heterogeneous chemistry through coating of particles with nitrates and sulphates (e.g., M7, Vignati et al., 2004). Some models also assume that freshly emitted dust can act as CCN, by assuming dust to be slightly hygroscopic (e.g., Oslo-Aero; Kirkevåg et al., 2018). Another mechanism by which dust can act as CCN is absorption of water vapour resulting in a surface film around the particle, known as absorption activation. Although there exist parametrisations that have been tested within ESMs (Karydis et al., 2017), most ESMs do not yet take this into account. Within mixed-phased and cirrus clouds regimes dust constitutes an important source of INP Froyd et al. (2022); Storelvmo (2017), however, ESMs often have a highly simplified way of treating INPs (Burrows et al., 2022). Typically, they parametrise the INP concentration as a function of temperature and humidity only (e.g., Meyers et al., 1992), which makes the models unable to represent changes to the INP concentration due to changes in dust concentration. In addition, a good representation of dust-cloud interactions is not only contingent on the inclusion of dust within the droplet activation scheme or ice nucleation scheme, but also requires

an accurate description of the physical properties of dust aerosols. Specifically, model assumptions about particle size and mineralogy also influence dust-cloud interactions. This is in part because the strength of cloud adjustments, resulting from dust radiative effects altering local thermodynamic conditions (often referred to as semidirect effects), depend on the levels of dust absorption and extinction in the model (Kok et al., 2023). Therefore, even for ESMs that include the representation of dust-cloud interaction either through CCN or INP, the accuracy of their representation is uncertain (Kok et al., 2023). Furthermore, these fundamental differences in the representation of dust-cloud interactions in ESMs might only have a limited impact on the net DuERF, as many of these interactions produce counteracting LW and SW radiative effects (McGraw et al., 2020).

Within the context of CMIP6, the *piClim-2xdust* experiment under AerChemMIP (Collins et al., 2017) is the most suitable modelling experiment to examine the climatic impact of a perturbation to the dust burden across different ESMs. The experiment initiates an idealised perturbation by scaling a suitable global dust emission tuning factor, internal to each model, such that, in principle, the dust emissions should be doubled. A total of nine different CMIP6 models participated in this experiment. We define DuERF as the difference in the TOA imbalance between *piClim-2xdust* and *piClim-control*, with the dust emission perturbation being the only factor that separates the two simulations. Although the relative increase in dust in the *piClim-2xdust* is comparable in magnitude to the estimated real world historical change, it is important to note the distinction between DuERF and dust effects diagnosed from this idealised setting and real-world historical dust forcing. Specifically, sea surface temperatures (SSTs) are fixed, anthropogenic aerosols are set to pre-industrial conditions, and the change in dust emission is imposed uniformly across dust source regions. Therefore, our findings cannot be directly compared with studies quantifying DuERF during the historical era (Leung et al., 2025). However, this idealised setting is still useful for investigating how ESMs behave in response to changes in dust burden. The DuERF results of the *piClim-2xdust* experiment published in Thornhill et al. (2021), based on six models (CNRM-ESM2-1, UKESM1-0-LL, MIROC6, NorESM2-LM, GFDL-ESM4 and GISS-E2), showed a weak multi-model mean DuERF of $-0.05 \pm 0.1$ W m$^{-2}$, see also Figure 1 b. This article expands on the results of Thornhill et al. (2021), by quantifying the direct and cloud DuERF in the ESMs, which was not done by Thornhill et al. (2021). We also examine how dust affects the flow of energy through the atmosphere and the impact of changes in the energy flow on global precipitation. We explain the differences in the models by examining intensive and extensive model properties associated with different aspects of the dust radiative effect, with a word of caution that not all required diagnostics are available in the standard CMIP6 model output. Extensive properties are referring to properties that depend on the amount of dust in the atmosphere, e.g., changes in cloud fraction, while intensive properties are model properties independent of the dust amount, e.g. dust optical properties. We use the insight on the relationship between DuERF and model parameters that regulate the dust forcing efficiency to argue that only perturbing the dust emission as in the *piClim-2xdust* experiment is insufficient to fully describe the uncertainty in DuERF and plead for a dust parameter perturbation experiment (PPE). PPEs have been used effectively to characterise uncertainty in aerosol forcing e.g. volcanic forcing, as demonstrated by (Marshall et al., 2019).

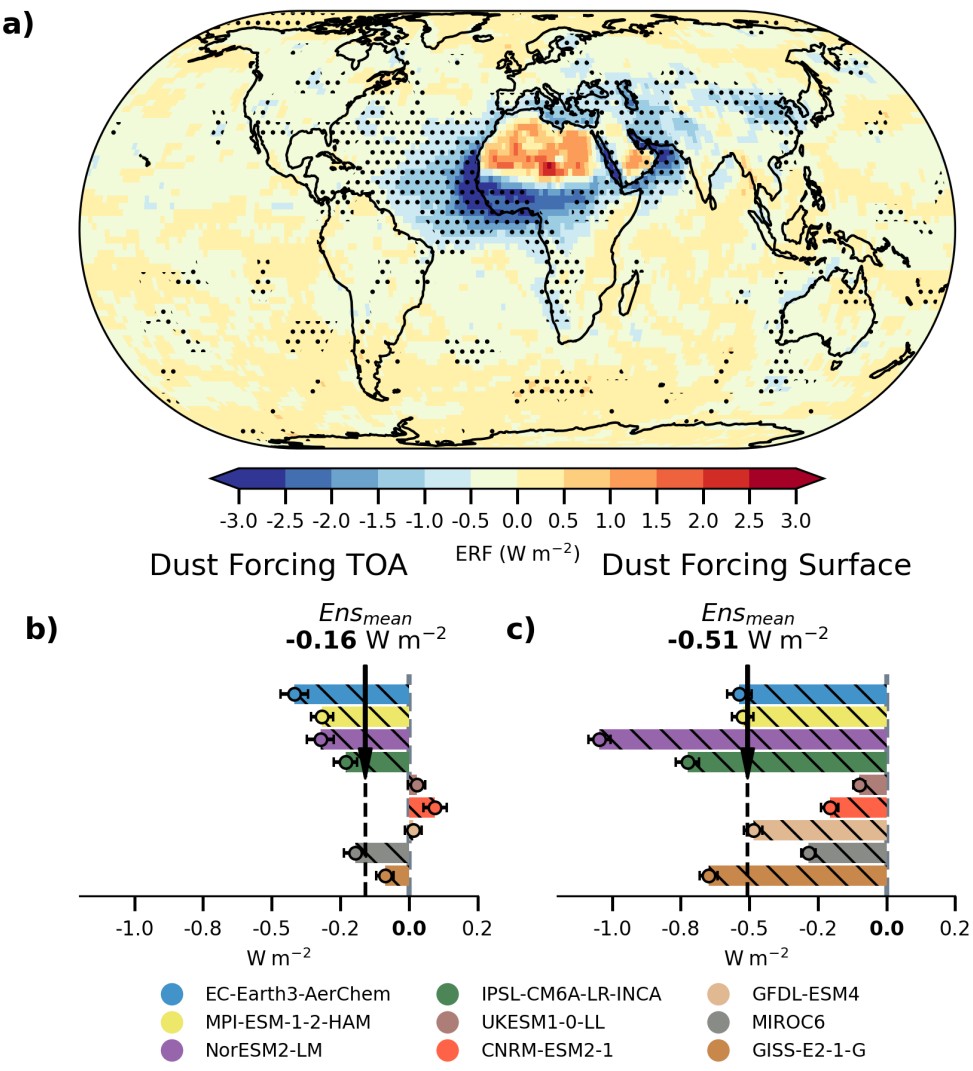

**Figure 1. (a)** Multi model mean DuERF from *piClim-2xdust* vs *piClim-control* alike Figure 1 in Thornhill et al. (2021). The stippling indicates where on the map at least 7 of the 9 models agree on the sign of the forcing. **(b)** Global mean DuERF for each model. **(c)** Global mean forcing at the surface. The error bar shows the standard error of the mean for each model.

## 2 Data and Methods

### 2.1 Description of CMIP6 experimental setup

The *piClim-2xdust* experiment belongs to the set of AerChemMIP perturbation experiments aimed at characterising the effective radiative forcing (ERF) of different climate agents, including the associated fast feedbacks (Collins et al., 2017). For this purpose, models participating in AerChemMIP are required to have an interactive aerosol scheme. The experimental design of the AerChemMIP ERF experiments uses fixed SSTs and sea ice area, prescribed at 1850 pre-industrial levels, consistent with the models' pre-industrial control simulation. Anthropogenic aerosol emissions and greenhouse gas concentrations are set at 1850 levels. The *piClim-2xdust* experiment doubles dust emissions by using a suitable tuning factor in the dust emission scheme of the model. Dynamical responses to such a dust perturbation may result in deviations from the expected doubling of emitted dust—this will be discussed in further detail later. Dust emission is calculated online driven by the surface wind speed. Additional factors such as the extent of bare soil, the texture of the soil, and the aridity of the surface also play critical roles in determining the dust source strength. After emission, dust is injected into the atmosphere with the models' assumptions on particle size distribution (see Table 1). Each model ran the simulation for at least thirty years to capture internal variability and give robust estimates of the simulated climatology. The setup of the reference simulation *piClim-control* is identical to *piClim-2xdust*, but with an unperturbed dust emission scaling factor. The difference between the two simulations is used to determine dust effects and DuERF in the different ESMs.

### 2.2 Model descriptions

In total, nine ESMs participated in the *piClim-2xdust* experiment. The AerChemMIP model data is provided open-access on Earth System Grid Federation (ESGF) data nodes. Table 1 provides an overview of the models used in this study, including specific model features that are relevant for the DuERF.

EC-Earth3-AerChem is specifically developed for AerChemMIP and includes interactive tropospheric aerosols and reactive greenhouse gases such as methane and ozone (van Noije et al., 2021). In this version, the standard EC-Earth3 (Döscher et al., 2022) is coupled to a chemical transport model, Tracer Model version 5 (TM5). TM5 operates on a coarser 3°x 2°horizontal grid with 34 levels, compared to the 80 km horizontal grid spacing with 91 vertical levels of the Integrated Forecast Model (IFS) cycle 36r4. Aerosol microphysics is simulated using the two-moment (number and mass) M7 scheme (Vignati et al., 2004), which is a modal scheme with four soluble modes and three insoluble modes. Mineral dust at emission is assigned only to the insoluble accumulation and coarse modes; however, dust can be transferred from the insoluble to the soluble modes via condensation of $H_2SO_4$ and by coagulation. The modes are described by lognormal distributions with fixed standard deviations. For effective refractive indices, dust is treated as internally mixed following the Maxwell-Garnett mixing rule. Furthermore, EC-Earth3-AerChem includes the absorption of LW radiation by mineral dust by using precomputed MACs.

MPI-ESM-1-2-HAM is the HAM (Hamburg Aerosol Module) version of the Max Planck Institute Earth System Model (MPI-ESM). The atmospheric component ECHAM6.3, is a spectral model. It uses version 2.3 of HAM and is detailed in Tegen et al. (2019). This version of HAM uses also the M7 modal aerosol scheme as EC-Earth3-AerChem. Similarly to EC-

Earth3-AerChem, dust is placed only in the insoluble modes and includes the same interactions between sulphate and mineral dust, which can transfer mineral dust from the insoluble to the soluble modes (Neubauer et al., 2019). HAM includes explicit calculations of cloud droplet and ice crystal number concentrations via a two-moment cloud microphysics scheme (Lohmann et al., 2007). Furthermore, mineral dust and black carbon particles can act as INPs, triggering heterogeneous ice nucleation.

The Norwegian Earth System Model, version 2 (NorESM2) (Seland et al., 2020), is a derivative of the Community Earth System Model (CESM), but it features an independent aerosol microphysical scheme known as Oslo-Aero (Kirkevåg et al., 2018). NorESM2 employs the Community Atmosphere Model version 6 (CAM6). Oslo-Aero is a modal aerosol scheme that utilises a 'production-tagged' approach, distinguishing it from other aerosol schemes by differentiating between background and process tracers. Process tracers, such as sulphate condensate and aqueous phase sulphate, act to modify the shape and chemical composition of the background modes, including the dust modes. When a process tracer is distributed within a background mode, it forms a mixture, and the composition of this mixture determines the optical properties of the background mode. Mineral dust is represented by two distinct background modes (number median radius of 0.22 $\mu$m and 0.62 $\mu$m), where 87% of the emitted mass is placed in the coarse mode. In addition to the solubility added by, for example, the condensing of sulphate on the dust aerosol, NorESM assumes dust to be slightly hygroscopic by default, which can make dust aerosols act as a potent CCN in the model (Kirkevåg et al., 2018). Furthermore, NorESM2 includes heterogeneous nucleation of ice by dust aerosols following classical nucleation theory (Hoose et al., 2010). However, the CMIP6 version of NorESM2 contained an error related to the ice limiter designed to ensure that the concentration of in-cloud ice did not exceed the available INPs. Unfortunately, the INPs calculated by the (Hoose et al., 2010) scheme were erroneously not included in this limit. Consequently, dust INPs in this model version can not contribute to the ice number within the mixed phase temperature regime (McGraw et al., 2023), but the scheme can still transform existing cloud droplets from liquid to ice; so, if dust leads to enhanced cloud droplet activation in the model, then cloud ice could be affected that way. NorESM2-LM has a separate scheme for heterogeneous nucleation via immersion freezing within cirrus clouds that is active and follows Liu et al. (2007).

The Institut Pierre Simon Laplace coupled model, version 6A (IPSL-CM6A-LR-INCA) uses the INteraction with Chemistry and Aerosols (INCA) aerosol module (Lurton et al., 2020) and the LMDZ6A dynamical core (Hourdin et al., 2020). The INCA model represents dust aerosols using a modal framework with one lognormal mode describing the dust aerosol size distribution, where each mode is treated as externally mixed (Balkanski et al., 2007). IPSL-CM6A-LR-INCA uses updated refractive indices for LW radiation interactions based on chamber measurements of Di Biagio et al. (2017, 2019). Dust aerosols are considered insoluble and do not act as CCN nor does the model represent dust as INP.

The UKESM1-0-LL model is developed by the UK Met Office and includes HadGEM3-GC3.1 as its dynamical core (Williams et al., 2018; Sellar et al., 2019). Unlike the modal representation of other aerosol species, dust aerosols are treated as an external mixture using a bin scheme. The six bin dust scheme (CLASSIC) has been found to produce reasonable results against present-day observed mass concentrations (Checa-Garcia et al., 2021). However, the separate treatment of the dust aerosols means that they do not act as CCN. UKESM1-0-LL does not either include a parametrisation of heterogeneous freezing with dust (Mulcahy et al., 2020).

The CNRM-ESM2-1 model, developed by CNRM-CERFACS, is based on version 6.3 of the ARPEGE-Climat model, which was originally derived from IFS (Séférian et al., 2019). Aerosols are simulated using the model's prognostic aerosol scheme, TACTIC_v2 (Tropospheric Aerosols for ClimaTe In CNRM-CM) (Michou et al., 2015), adapted from the IFS scheme. TACTIC_v2 includes 12 prognostic aerosol variables. Dust is represented using a sectional model with three size bins, and its optical properties are fixed. Dust is not considered to act as CCN or INP in the model. CNRM-ESM2-1 includes interactions between vegetation and dust, using interactive aerosols and chemistry to simulate feedbacks and interactions between dust emissions and changes in vegetation and land cover.

The Model for Interdisciplinary Research on Climate version 6 (MIROC6) is developed by a Japanese modelling consortium (Tatebe et al., 2019). MIROC6 uses a spectral dynamical core and employs the Spectral Radiation Transport Model for Aerosol Species (SPRINTARS) aerosol scheme. Dust is represented by a sectional scheme with six bins ranging from 0.2 to 10.0 $\mu m$ in particle radius. SPRINTARS includes microphysical parametrisations of dust-cloud interactions for both ice and liquid clouds (Takemura et al., 2009). The heterogeneous nucleation of the ice is based on a formulation similar to that of MPI-ESM-1-2-HAM (Lohmann and Diehl, 2006). Dust is considered to be a CCN by assuming the dust aerosols to be slightly hygroscopic, similar to NorESM2-LM. Dust aerosols are treated as externally mixed and therefore do not interact chemically with other trace species in the model.

The GISS-E2-1-G model is developed by the NASA Goddard Institute for Space Studies. The AerChemMIP configuration of the model includes the One-Moment Aerosol (OMA) module. OMA is a mass-based aerosol scheme with prescribed aerosol sizes and properties. Furthermore, aerosols are treated as externally mixed, except for dust and sea salt. Dust aerosols are represented using five size bins ranging from 0.1 to 16 $\mu m$ in particle radius (Bauer et al., 2007). Dust aerosols do not directly impact cloud droplet concentration because dust is not included in the hygroscopic mass fraction of aerosols that can participate in cloud nucleation processes (Schmidt et al., 2014). However, dust can be coated with sulphate and nitrate, allowing dust to act as a sink for other CCNs. GISS-E2-1-G does not simulate heterogeneous ice nucleation and therefore does not include dust aerosols as INPs.

## 2.3 Diagnosing simulated changes due to increased dust

To diagnose the dust-induced changes in the models from the *piClim-2xdust* experiment, we take the climatology of *piClim-2xdust* and subtract the climatology of *piClim-control*, with the latter being the corresponding control experiment without any perturbations. Since there are no other changes to the model, we assume that the difference in a given model output diagnostic is due to dust-induced effects. For the *piClim-2xdust* experiment we discard the first year to allow the model to spin up properly, otherwise the climatologies are calculated by first resampling the model output into annual averages and then averaging over all the model years. To determine if the dust-induced effects are significant, we test the following hypothesis, using a two-sided t-test, again on annual data:

$$\text{H}_0: \quad \text{There is no change in climatology in the model;} \quad \mu_{2xdust} - \mu_{control} = 0 \tag{1}$$

$$\text{H}_\text{A}: \quad \text{The dust perturbation changes the climatology;} \quad |\mu_{2xdust} - \mu_{control}| > 0. \tag{2}$$

The statistic of the t-test is calculated by first finding the pooled standard deviation of the 30 (29) year mean of the *piClim-control* (*piClim-2xdust*) in order to account for the two simulations having different variances. The pooled standard deviation is calculated using Equation 3:

$$\sigma_{\overline{X}2xdust - \overline{X}ctrl} = \sqrt{\frac{(N_{2xdust} - 1)\sigma^2_{\overline{X}2xdust} + (N_{ctrl} - 1)\sigma^2_{\overline{X}ctrl}}{N_{2xdust} + N_{ctrl} - 2}}, \tag{3}$$

Where $N_{2xdust}$ and $N_{ctrl}$ are the numbers of simulated years included for the *piClim-2xdust* and *piClim-control* simulations, respectively. $\overline{X}$ signifies the average of a given diagnostic. The pooled standard deviation is then used to calculate the standard error, $s_{\overline{X}_{2xdust} - \overline{X}_{ctrl}}$, which is subsequently used to calculate the test statistic for the t-test:

$$t = \frac{\overline{X}_{2xdust} - \overline{X}_{ctrl}}{s_{\overline{X}_{2xdust} - \overline{X}_{ctrl}}}. \tag{4}$$

To determine significance, the computed t-statistic is compared with the critical t-value at the 0.05 significance level for a
235 two-tailed test.

## 2.4 Dust Forcing decomposition

To decompose the DuERF we use the standard method of Ghan (2013). The Ghan decomposition requires the so called 'aerosol-free' diagnostics, that comes from an additional call to the radiation code where the scattering and absorption by aerosols are set to zero. Seven of the nine models (see Table 1) provided these diagnostics. The DuERF is defined as the difference in
the TOA imbalance between *piClim-control* and *piClim-2xdust*, and is decomposed into direct and cloud DuERF following Equations 5–8:

$$\text{DuERF} = \Delta\text{F} \Rightarrow \Delta(\text{rsut} + \text{rlut} - \text{rsdt}) \tag{5}$$

$$\text{Direct DuERF} = \Delta\left(\text{F} - \text{F}_{\text{clean}}\right) \Rightarrow \text{DuERF} - \Delta(\text{rsutaf} + \text{rlutaf} - \text{rsdt}) \tag{6}$$

$$\text{Cloud DuERF} = \Delta\left(\text{F}_{\text{clean}} - \text{F}_{\text{clear,clean}}\right) \Rightarrow \Delta(\text{rsutaf} + \text{rlutaf} - \text{rsdt}) - \Delta(\text{rsutcsaf} + \text{rlutcsaf} - \text{rsdt}) \tag{7}$$

$$\text{Albedo DuERF} = \Delta\text{F}_{\text{clear,clean}} \Rightarrow \Delta(\text{rsutcsaf} + \text{rlutcsaf} - \text{rsdt}) \tag{8}$$

The F, $F_{clean}$ and $F_{clear,clean}$ is the TOA forcing of all-sky, all-sky aerosol-free and clear-sky aerosol-free, respectively. The variables after the arrow refer to the names of the CMOR diagnostics actually used. The $\Delta$ symbol implies the difference between *piClim-2xdust* and *piClim-control*. To obtain the direct ERF, we subtract the aerosol-free fluxes from the DuERF, thereby eliminating the ERF through cloud and surface albedo changes. Similarly, to calculate the cloud DuERF, we subtract
clear-sky aerosol-free fluxes from the aerosol-free fluxes. The cloud DuERF includes the radiative impacts of cloud adjustments on changes in the thermal structure of the atmosphere (both in-direct and semi-direct effects).

## 2.5 Top-Down energy view on dust-driven precipitation changes

The energetic perspective provides a 'top-down' approach to examine the effects of aerosols on precipitation, bypassing some of the complexities associated with poorly resolved and diagnosed microphysical processes. Instead, it relies on thermodynamic

processes, which are typically well represented in ESMs. In case of radiative equilibrium (Eq. 9), global precipitation is generally governed by the balance between latent heat release ($L$), sensible heat flux ($H$) and atmospheric radiative cooling (ARC) (Zhang et al., 2021; Pendergrass and Hartmann, 2014). ARC is defined as the difference between the net LW and SW fluxes at TOA and the surface. Latent heat is proportional to precipitation and represents approximately two-thirds of the net sensible plus latent energy flux, therefore, there is a strong correlation between ARC and precipitation (Stephens et al., 2012). Since SSTs are fixed in the *piClim* experiments, these experiments do not include temperature-driven responses of dust on global precipitation, which is mainly determined by TOA forcing. Consequently, the precipitation response should be interpreted as a fast response.

$$\overbrace{\Delta F_{TOA} - \Delta F_{Srf}}^{\text{ARC}} + \Delta L + \Delta H = 0. \tag{9}$$

The fast response scales with the change in ARC. Scattering aerosols do not affect the ARC because the increase in SW flux at the TOA equals the reduction in SW flux at the surface, and thus the ARC remains unchanged. In contrast, absorbing aerosols (e.g., certain types of dust minerals) reduce the net radiative flux more at the surface than they outgoing SW flux at the TOA, leading to a positive ARC. As a result, the sum of $\Delta L$ and $\Delta H$ must be negative for the balance to hold, and thus precipitation decreases. Furthermore, since dust also acts as INPs, dust can increase the ice-cloud fraction, which reduces the outgoing TOA LW flux, which would also lead to a positive ARC. The physical interpretation is that atmospheric heating above a surface with a constant temperature increases atmospheric stability due to a reduced lapse rate, which in turn weakens convection.

## 3 Results

### 3.1 Spatial Distribution and Model Variability of DuERF

The multi-model mean DuERF from the nine models is shown in Figure 1a. DuERF has the largest negative values above the areas where dust blows out over the ocean. Furthermore, all models consistently show a stark land-ocean contrast in the spatial pattern of DuERF, with some models exhibiting a change of sign in the DuERF in the transition from ocean to land areas (Figure S10). In NorESM2-LM, EC-Earth3-AerChem and MPI-ESM-1-2-HAM, the discontinuity between ocean and desert is less pronounced and the sign is not reversed, as is the case for CNRM-ESM2-1, IPSL-CM6A-LR-INCA, and UKESM1-0-LL (Supplement Figure S2–S3). However, in terms of the albedo of the desert surface, the models are relatively consistent (Supplement Figure S1), suggesting that the model-spread in forcing efficiencies above deserts is largely driven by model differences in intensive dust properties. Intensive properties such as MAC, the fraction of coarse-mode dust, and the height of dust in the upper troposphere all contribute to local heating (Claquin et al., 1998), while the dust SSA governs the cooling effect. Together, this determines the surface albedo threshold from where the forcing switches from negative to positive. Satellite observations show that there is little contrast between dust and the desert surface below; therefore, the forcing per unit of DOD

should be close to zero (Patadia et al., 2009) above the desert. This is incongruous with the high positive forcing observed in several of the ESMs (Figures S2 and S10). The interaction between dust's intensive properties and surface characteristics plays a crucial role in determining the dust radiative effect above desert regions in the ESMs. Therefore, updates to the dust composition are suggested to be accompanied with updates to the desert surface albedo to avoid biases in the dust direct forcing efficiency due to inconsistencies between the optical properties of the dust and the desert surface.

In regard to the global mean forcing DuERF shown in Figure 1b the 30-year simulation length appears to be adequate to obtain a representative estimate of DuERF, with standard errors of less than $0.1\,\mathrm{Wm}^{-2}$ for most models. The inclusion of additional models beyond those used by Thornhill et al. (2021) has increased the simulated range of DuERF, with our model ensemble showing a range from $0.09\,\mathrm{Wm}^{-2}$ to $-0.41\,\mathrm{Wm}^{-2}$ compared to $0.09\,\mathrm{Wm}^{-2}$ to $-0.18\,\mathrm{Wm}^{-2}$ reported in Thornhill et al. (2021). The increased range of DuERF reflects the addition of MPI-ESM-1-2-HAM and EC-Earth3-AerChem, which are models that exhibit a large negative DuERF. Furthermore, CNRM-ESM2-1 stands out as the only model that has a significant positive DuERF, while UKESM1-0-LL and GFDL-ESM4 show a small positive mean DuERF, their standard error indicating that it is not significantly different from zero. The other six models all show negative DuERF, which leads to a more negative ensemble mean DuERF of $-0.16\,\mathrm{Wm}^{-2}$ compared to $-0.05\,\mathrm{Wm}^{-2}$ reported in Thornhill et al. (2021).

Although this study examines DuERF from a global angle, note that the models also differ substantially in their regional distribution of dust source regions (Supplement Figure S4). In particular, they disagree on the relative importance of East Asian dust sources. Such dust source differences would likely contribute to the inter-model spread in the DuERF since different regions bring into play different forcing efficiencies. Addressing this question would require prescribing the dust in the ESMs with a consistent dust emission inventory (e.g., Leung et al., 2025) as a sensitivity study.

The DuERF at the surface is disproportionate to the TOA DuERF (Figure 1c). This discrepancy is the smallest in EC-Earth3-AerChem, MIROC6 and MPI-ESM-1-2-HAM. In the other models, the surface forcing in absolute terms is between 2–6 times larger than at TOA. Moreover, in UKESM1-0-LL, CNRM-ESM2-1, and GFDL-ESM4, net forcing changes from positive at TOA to negative at the surface. The imbalance between the surface and TOA implies that additional energy is absorbed in the atmosphere, hence this additional energy has to be balanced by a reduction in latent and sensible heat fluxes (Eq. 9).

### 3.2 Impact of extensive and intensive dust properties on modelled dust direct ERF

In this section, we examine the direct DuERF from the AerChemMIP models (Figure 2) and how differences in the direct DuERF are tied to model differences in dust intensive and extensive properties. Direct DuERF is only provided for the models that provided the required aerosol-free diagnostics (see Table 1). Figure 2a shows that in this subset of seven models the modelled range of net direct DuERF spans from $-0.56$ to $+0.05\,\mathrm{Wm}^{-2}$, with the SW component ranging from $-0.68$ to $+0.025\,\mathrm{Wm}^{-2}$, and the LW component varying between $+0.01$ and $+0.19\,\mathrm{Wm}^{-2}$. To put the ERF from the *piClim-2xdust* experiment into context, the multi-model mean direct DuERF is comparable to the direct radiative forcing due to anthropogenic sulphate aerosol (Kalisoras et al., 2024).

It is interesting to compare our direct DuERF values and range with other estimates of the dust effective radiative effect (DuERE). As discussed below, doubling the global dust tuning constant did not always lead to a 100% increase in dust emis-

sions. Therefore, by scaling our DuERF values, we correct for this and arrive at an estimate of the pre-industrial DuERE
(Figure S5). These direct DuERE values of the ESMs (Figure S5) generally align with the Kok et al. (2023) assessed range
for a direct DuERE of $-0.5$–$0.2\,\mathrm{Wm^{-2}}$, except EC-Earth3-AerChem, which exhibits a DuERE that is more negative than
this range. Regarding the LW direct DuERE, EC-Earth3-AerChem, NorESM2-LM, and CNRM-ESM2-1 all exhibit LW direct
DuERE values that are one order of magnitude smaller than the assessed range of $+0.1$ to $+0.4\,\mathrm{Wm^{-2}}$ reported by Kok et al.
(2017, 2023) (Figure S5). Although the ESMs exhibit SW direct DuERE values that are generally better aligned with the as-
sessed range of $-0.1$ to $-0.7\,\mathrm{Wm^{-2}}$ (Kok et al., 2023), CNRM-ESM2-1 falls outside this range by exhibiting a positive SW
direct DuERE.

The dust direct forcing efficiency is shown in Figure 2b. Removing the influence due to differences in the change in DOD
between *piClim-2xdust* and *piClim-control* among the models makes the models appear more coherent. In all models except
UKESM1-0-LL, the LW forcing efficiency in absolute value is about an order of magnitude lower than the SW forcing effi-
ciency, implying that models are largely unable to represent LW scattering from the coarse to super-coarse dust particles. With
the exception of GFDL-ESM4 and CNRM-ESM2-1, the SW forcing efficiency is relatively similar between the models. Since
the LW forcing efficiency is minor, the proportion of SW absorption to total extinction or SSA of the dust in the models appears
to largely determine the dust forcing efficiency.

For the surface forcing efficiency, we use the change in surface clear sky fluxes as the dust direct surface forcing (which could
be calculated for all nine models). We see that quite some models with small direct DuERF show a disproportional efficient
reduction in radiation at the surface, e.g., CNRM-ESM2-1 and GFDL-ESM4. Furthermore, several models also show a large
discrepancy between the SW and net clear-sky forcing efficiency, e.g., UKESM1-0-LL and CNRM-ESM2-1. This implies a
positive LW clear-sky effect on the surface, by (1) LW backscatter to the surface by coarse dust or (2) dust SW absorption
heating the atmosphere and thus increasing emission of LW radiation back towards the surface. In EC-Earth3-AerChem, MPI-
ESM-HAM-1-2 and NorESM2-LM, we can clearly see that SW clear-sky forcing explains most of the net surface clear-sky
forcing.

We further examine how much the 2xdust perturbation translates into global mean changes in dust emission, burden, dust
optical depth (DOD), and dust absorption optical depth (DAOD) and how the inter-model differences relate to the intensive dust
characteristics of the models such as the mass extinction coefficient (MEC), mass absorption coefficient (MAC), lifetime, dust
Angstrom exponent, and fraction of wet to total deposition (Figure 2c). We define DOD (DAOD) as the change in the optical
depth diagnostic variable of total aerosol (absorption) from *piClim-2xdust* to *piClim-control*, as dust-exclusive aerosol optical
depth diagnostics were not available for some ESMs. For the extensive dust properties in Figure 2c, the changes relative to
*piClim-control* are shown in parentheses. The multi-model data are displayed in a heatmap, where the most intensely coloured
green represents the model that ranks highest within each column (dust cycle/optical parameter). Any gaps in the table denote
instances where the models did not provide the requested variable. The final row of the table contains the multi-model mean.

The absolute change in emitted dust varies significantly between the models, largely because of the vastly different as-
sumptions regarding the dust particle size distribution. The amount of the added dust emissions differs by almost an order of
magnitude, with EC-Earth3-AerChem showing the smallest increase (956 Tg year$^{-1}$) and UKESM1-0-LL showing the largest

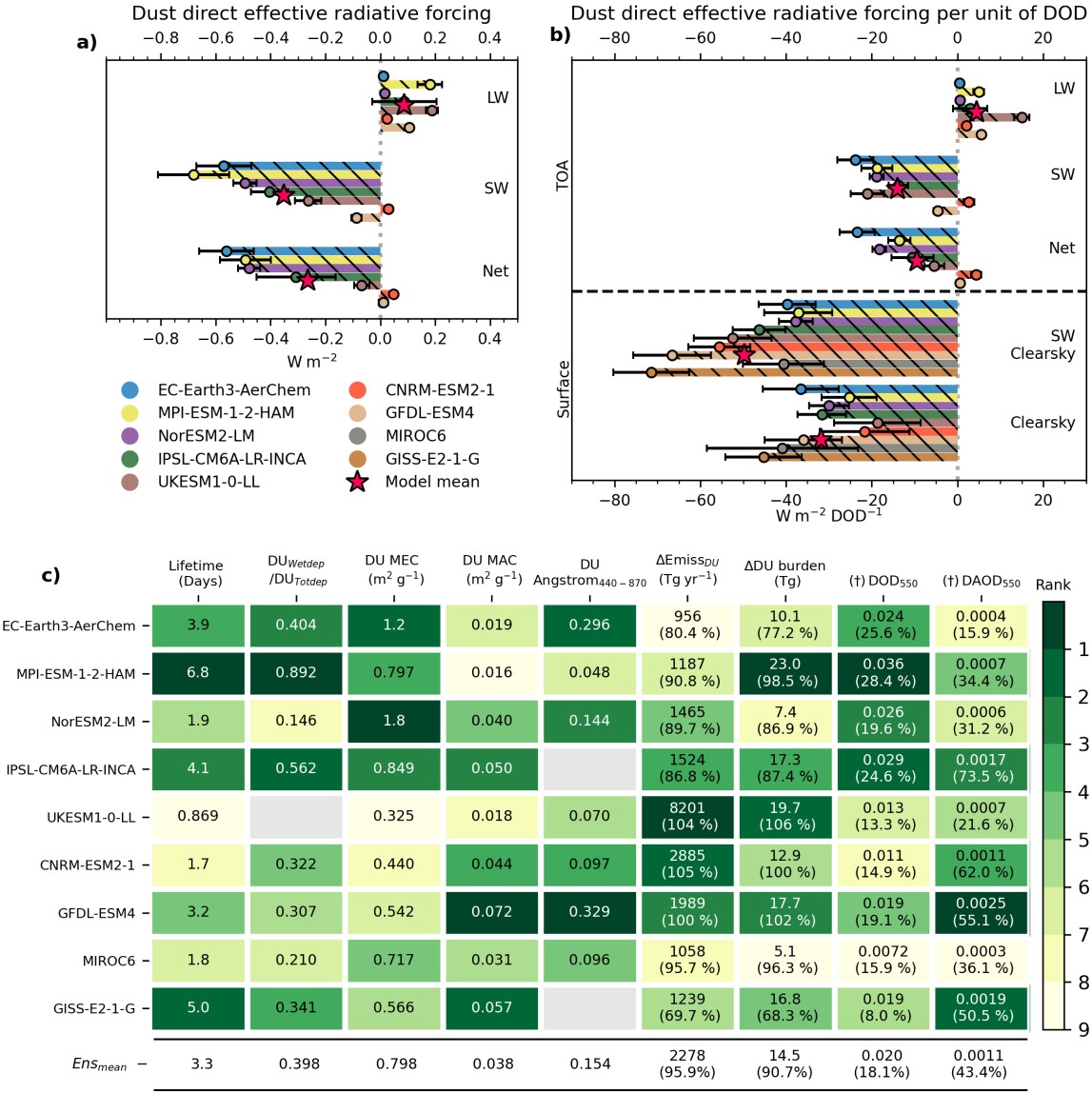

**Figure 2.** Global mean dust direct effective radiative forcing (**a**) and direct effective forcing efficiency (**b**) from *piClim-2xdust* vs *piClim-control*. The forcing efficiency is shown for both the surface and TOA, while the radiative forcing is only for TOA. For each model the error-bar indicates the model's standard error of the mean forcing. The red star indicates the multi-model mean. Global mean diagnostics of dust cycle and optical parameters (**c**) are presented. Intensive parameters ($DU_{Wetdep}/DU_{Totdep}$, Lifetime, $\text{Angstrom}_{440-870}$, DU MAC and DU MEC), are exclusively related to dust representation in the model. Dust Angstrom coefficient is calculated based on the change in AOD440 and AOD870. The dust mass extinction (absorption) coefficient DU MEC (DU MAC) is defined as $DOD_{550}$ ($\Delta DOD_{550}$) divided by $\Delta DU$ burden. Lifetime is approximated as $\Delta DU$ burden divided by $\Delta DU$ Totdep. Extensive parameters dependent on dust load ($\Delta Emiss_{DU}$, $\Delta DU$ burden, $DOD_{550}$, $DOD_{550}$) are depicted as the differences between *piClim-2xdust* and *piClim-control*, with the corresponding relative changes from *piClim-2xdust* indicated in parentheses. The shading shows the ranking of the models for a given diagnostic, from the model with the largest value (dark-shading) to the model with the smallest value (light shading).

increase (8262 Tg year$^{-1}$) (Figure 2c). Most of the models exhibit an increase in the emitted dust mass between 1000 and 2000 Tg year$^{-1}$. The experiment setup of doubling the dust emissions implies that this added emitted dust should be approximately the amount of dust emitted in the reference model. However, the increase in dust emission relative to *piClim-control*, is about 96% for the multi-model mean. Furthermore, there is considerable variability among the models; for instance, GISS-E2-1-G achieved only a 70% increase, while CNRM-ESM2-1 exhibited the largest increase at 105%. Such substantial inter-model differences in the relative increase in emissions in an experiment designed to invoke a doubling (100% increase) is somewhat surprising, possibly pointing to dynamical feedbacks of added dust on dust source strength itself. However, for our purpose of decomposing forcing and understanding inter-model variability, this is not too important, since we analyse the forcing and properties of the added dust. Differences in just the relative increase in emission strength between models do not explain the magnitude of the inter-model differences in the direct DuERF.

In six of the nine models, dry deposition is the dominant removal mechanism. Dry deposition is the most efficient for removing coarse to super-coarse dust particles. Models that exhibit a predominate role of dry deposition tend to correlate with shorter dust lifetimes and often include a larger fraction of super-coarse dust. Only IPSL-CM6A-LR-INCA and MPI-ESM-1-2-HAM have wet deposition as the main removal process. A predominant role of wet deposition tends to correlate with longer dust lifetimes (columns 2–3 Figure 2c), given that dust that is not removed by dry deposition close to the source will eventually be removed by wet deposition far from the source. The global dust load in the models is determined by the balance between emission strength and removal efficiency, where models with high emissions (UKESM1-0-LL) or a large fraction of wet deposition, and thus a small fraction of dry deposition close to the source (MPI-ESM-1-2-HAM) typically have the highest dust loads. The removal processes thus significantly affect the burden ranking of the models, where models with lower emissions can still exhibit high dust burdens. This shows that altering the dust emission strength is not the sole parameter in the dust cycle that could impact the DuERF.

The change in annual mean DOD and DAOD over that from *piClim-control* for the 9-model ensemble is $0.0204 \pm 0.009$ and $0.0011 \pm 0.0008$, respectively. This change equates to a relative increase in total AOD between 8–28% and AAOD between 16–74% compared to *piClim-control* —the relative change is less than 100% since AOD and AAOD include more aerosol species than dust alone. The resulting changes in DOD and DAOD in response to a disturbance in the global dust burden depend upon DU MEC and DU MAC in the model. Models with large DU MEC and DU MAC can compensate for low burdens and may exhibit high DOD. This effect is illustrated by NorESM2-LM and EC-Earth3-AerChem, which have low dust loads (7.4 Tg and 10.1 Tg, respectively), but have a larger dust MEC, resulting in a relatively large DOD (0.026 and 0.024, respectively). Most models align on the increase in DOD, and the majority of models indicate changes ranging from 0.02 to 0.04, closely matching the uncertainty range in the present-day DOD reported by Ridley et al. (2016). This demonstrates how emissions, removal efficiency, and extinction coefficients are possibly tuned in the models to ensure a reasonable DOD in the unperturbed baseline. For models with a large DU MAC, DAOD can be responsible for up to 70% of total AAOD. In these models, absorption can account for between 6–13% of the DOD. In contrast, in models with weakly absorbing dust, such as EC-Earth3-AerChem, MPI-ESM-1-2-HAM, and UKESM1-0-LL, absorption only accounts for between 0.02–2% of DOD.

The most direct link we find between direct DuERF and the dust cycle and dust optical properties is related to DAOD and DOD. The amount of absorption and total extinction in the model together explain quite a large part (88%) of the inter-model variation in the total direct DuERF (supplement Figure S6), where models with a low DOD and a larger DAOD exhibit a smaller negative if not positive direct DuERF and vice versa.

Overall, the AerChemMIP ensemble mean indicates a negative net direct DuERF of $-0.25$ W m$^{-2}$ or a forcing efficiency of $-10$ W m$^{-2}$ per unit of optical depth. We caution that accounting for LW scattering and underestimation of super-coarse dust could still alter these results, but it is not possible to diagnose the LW effects from the standard output. Despite its simple design, the *piClim-2xdust* experiment appears to give quite complex results, as demonstrated by the few key dust diagnostics selected and shown in Figure 2c. This complexity is apparent in how the models can be relatively consistent in the global mean DOD, a quantity that is generally well constrained by satellite observations, while using substantially different frameworks to represent the dust cycle. This shows that constraining DOD alone is not sufficient to reduce the uncertainty in direct DuERF. Going forward, we need to expose ESMs to a larger set of constraints on different aspects of the dust cycle, for example, particle size distribution (Kok et al., 2021), CRI (Li et al., 2024; Wang et al., 2024), or spatial gradients in DOD to constrain the lifetime of dust to reduce the uncertainty in direct DuERF.

### 3.3 Dust cloud forcing and changes in associated cloud characteristics

Dust causes radiative perturbations via clouds by modifying the thermodynamic environment and by serving as CCN and INPs. The cloud DuERF is determined by the extent of the dust perturbation and the amount of pre-existing dust, and as this relationship is non-linear, we refrain from retrieving an effective forcing efficiency of dust-cloud interactions from the *piClim-2xdust* experiment analysed here. In the following section, we examine the cloud DuERF and associated changed cloud characteristics across the AerChemMIP ESMs.

Figure 3a shows the LW, SW and net cloud DuERF. For LW cloud DuERF, all models, except NorESM2-LM, display a slightly negative forcing, ranging from $-0.1$ to $0.0$ W m$^{-2}$. In contrast, NorESM2-LM shows a substantial positive LW cloud DuERF of $0.66$ W m$^{-2}$, resulting in a slightly positive multi-model mean LW cloud DuERF. Regarding the SW cloud DuERF, NorESM2-LM again diverges with a substantial negative forcing of $-0.56$ W m$^{-2}$. Among the other models, most show a positive SW cloud DuERF, ranging from $-0.03$ to $0.23$ W m$^{-2}$. Despite the notable differences in the sign and magnitude of individual LW and SW components of the cloud DuERF between NorESM2-LM and other models, there is more agreement on the total cloud DuERF, which ranges from $-0.04$ to $0.16$ Wm$^{-2}$. To understand why the cloud DuERF in NorESM2-LM differs significantly from other models, we investigate simulated changes in cloud characteristics (Figure 3b). Notably, NorESM2-LM uniquely shows a significant increase in both the ice water path (IWP) and the high cloud fraction predominately at temperatures below $-37\,^{\circ}$C (Figure S9), consistent with the increase of dust INPs enhancing cirrus cloud lifetimes and thus amount. Cirrus clouds are characterised by competition between homogeneous freezing and deposition ice nucleation (Burrows et al., 2022), where elevated INP concentrations can decrease the cloud ice particle number concentration by promoting the growth of larger ice particles, which consume the supersaturation required for homogeneous freezing, thus inhibiting the formation of smaller, longer-lived ice crystals (Storelvmo, 2017). However, in regions where heterogeneous ice nucleation predominates, additional

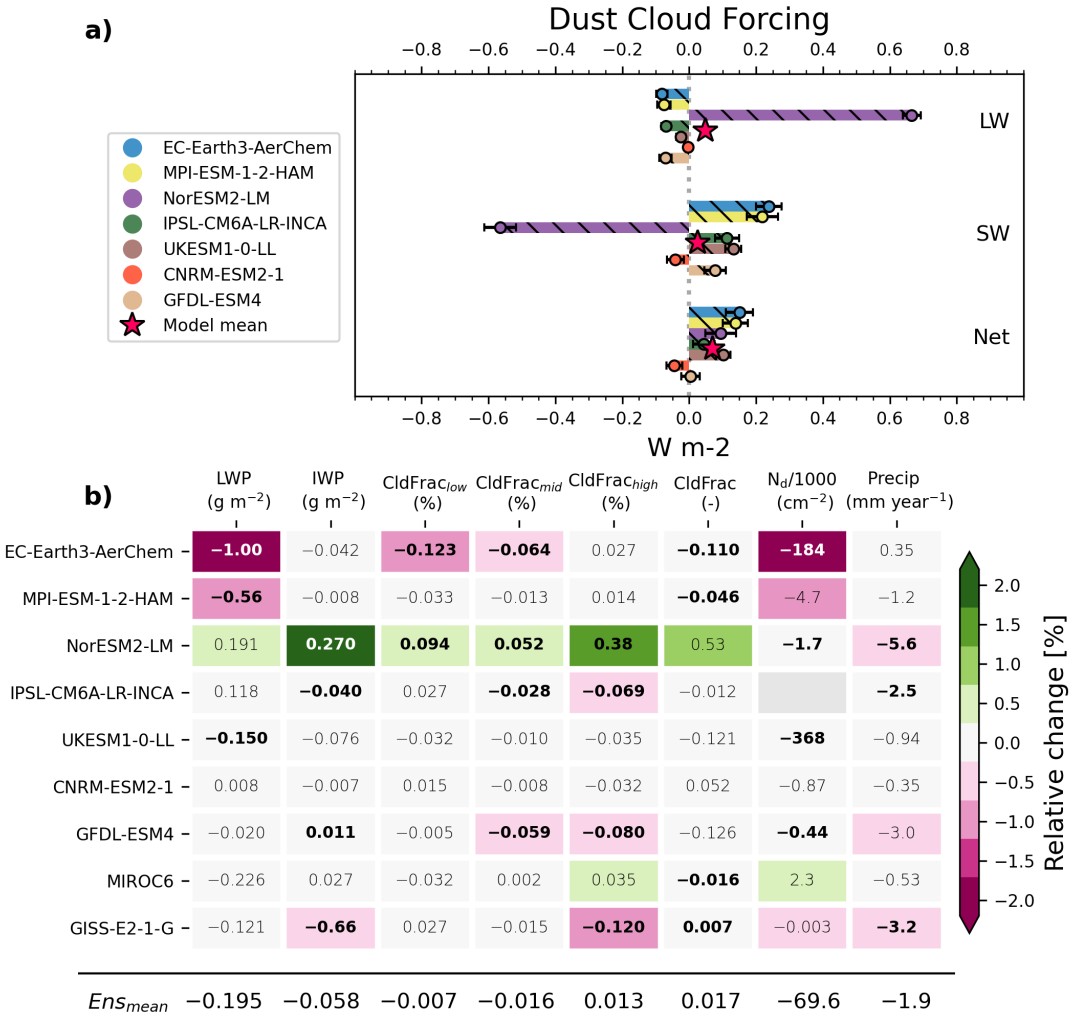

**Figure 3.** (**a**) Global mean cloud dust effective radiative forcing (cloud DuERF). The error bars correspond to one standard deviation of the modelled cloud DuERF and the red stars indicate the multi-model mean. (**b**) Global mean change due to dust (*piClim-2xdust - piClim-control*) of the following cloud properties: liquid water path (LWP), ice water path (IWP), low, medium and high and total cloud fraction (CldFrac), cloud droplet number concentration (Nd), precipitation (Precip). Bold values indicate that the difference between *piClim-2xdust* and *piClim-control* is significantly different from zero at a 95% confidence level. The colour shading shows the relative change between the two simulations.

INPs typically increase ice crystal concentrations (Storelvmo, 2017), which appears to characterise NorESM2-LM. Note that due to a known bug (McGraw et al., 2023), heterogeneous ice nucleation can only change cloud ice particle number within the cirrus regime in NorESM2-LM.

In contrast to NorESM2-LM, MPI-ESM-1-2-HAM, which also includes an aerosol-aware INP scheme, shows no significant changes in IWP or high cloud fraction, resulting in a near-zero LW cloud DuERF. This aligns with Dietlicher et al. (2019), where the ice formation within mixed-phased clouds in ECHAM6.3-HAM (the atmospheric model of MPI-ESM-1-2-HAM), is mainly dominated by homogeneous freezing, with contact and immersion freezing contributing only 6% to cloud ice formation. Furthermore, in general, ECHAM6.3 has been shown to be largely intensive to perturbations in heterogenous freezing processes
(Proske et al., 2023).

Consequently, NorESM2-LM stands out as the only model within the AerChemMIP ensemble displaying a notable dust impact on cirrus clouds. This raises questions about whether it is an outlier or if similar behaviours would emerge if more models adopt aerosol-aware INP representations. Regardless, the observational evidence shows that the role of dust as an INP is an ubiquitous part of cirrus cloud formation, supporting the response observed in NorESM2-LM (Froyd et al., 2022).

Next, we examine models that lack an aerosol-aware INP representation or are not sensitive to dust INPs, including EC-Earth3-AerChem, MPI-ESM-1-2-HAM, IPSL-CM6A-LR-INCA, UKESM1-0-LL, and GFDL-ESM4. These models commonly employ INP representations that are based on empirical relationships among humidity, temperature, and INP concentration (Burrows et al., 2022). Dust perturbations can indirectly influence cloud ice fraction by altering atmospheric temperature and humidity, however, as shown by the generally insignificant changes in IWP and high cloud fraction, this effect
is minor (Figure 3b). Also, in ESMs that show a significant, albeit small, change in the high cloud fraction (IPSL-CM6A-LR-INCA,GFDL-ESM4 and GISS-E2-1-G), the high-cloud fraction is reduced. In this case, we interpret this reduction to be caused by the added dust absorption weakening the deep convection, as has been suggested also by (Jiang et al., 2018) as possible effect of dust. Note, that the relative increase in AAOD in these models was 50% or higher.

With regard to dust impacts on liquid clouds, we observe that EC-Earth-AerChem and MPI-ESM-1-2-HAM have the largest
relative decrease in Nd. These two models share the same aerosol microphysical scheme (Table 1) and do not consider freshly emitted dust to be a CCN, dust must first undergo chemical ageing. Here, more dust would increase the surface area available for the condensation of aerosol precursors (e.g., $SO_2$), thus there would be less available to form secondary aerosols and possibly less CCN available. The decrease in Nd could also be a response to reduced evaporation and cloud cover, driven by the dust surface cooling. However, unfortunately the CCN diagnostics were generally not provided by the models (Supplement
Figure S7), therefore, we can only offer our hypothesis but not rigorously test it. However, comparing the CCN changes between NorESM2-LM and MPI-ESM-1-2-HAM supports this interpretation (Supplement Figure S7). The models with least SW cloud DuERF are also the models with more absorbing dust, such as GFDL-ESM4 and IPSL-CM6A-LR-INCA. Absorbing aerosols can increase the temperature in the atmospheric layer above the cloud, causing increased stability and enhancing the cloud cover. This stabilisation acts as a semidirect negative cloud DuERF. However, positive dust semidirect effects also exist,
where dust that resides within the cloud would act to decrease cloud cover through enhanced cloud evaporation. However, to

disentangle the impact of the vertical distribution of dust on clouds requires collocating the dust mass mixing ratio with the cloud fraction on a high temporal frequency, output that is not currently available in the models.

Contrasting direct DuERF (Figure 2a) and cloud DuERF (Figure 3 a), we see that the inter-model spread and magnitude of DuERF are dominated by direct DuERF. However, the larger spread in direct DuERF should not be interpreted as the cloud DuERF being less uncertain compared to direct DuERF, as current ESMs cannot be trusted to accurately depict the uncertainty in dust-cloud interactions. This only shows that the ESMs currently have larger diversity in how they represent direct radiative effects of dust compared to indirect radiative effects. Given that most ESMs lack crucial processes for depicting dust-cloud radiative effects, e.g., aerosol-aware INP representation, the apparent model consistency is due to a lack of representation and not lack of uncertainty. The DuERF is also different from the anthropogenic aerosol ERF (e.g., IPCC AR6, Forster et al., 2021), which shows that aerosol indirect forcing is the largest and most uncertain aspect of aerosol radiative forcing. However, the dust radiative effect is in several aspects different from the indirect effect of soluble aerosols; for example, dust influences both liquid and ice clouds, and the SW and LW radiative effects can pull in opposite directions (McGraw et al., 2020), making the overall dust cloud radiative effect appear weaker than that of anthropogenic aerosols.

The Ghan (2013) decomposition includes a 'residual' term that is attributed to changes in albedo (Figure S12). With respect to the global mean value, the albedo DuERF ranges from $-0.01$ to $0.14\ \mathrm{W\ m^{-2}}$, and except for NorESM2-LM and CNRM-ESM2-1, it is below $0.05\ \mathrm{W\ m^{-2}}$. The spatial distribution of the albedo forcing is also not consistent between the ESMs. Consequently, we provide the albedo term for completeness of the decomposition in the supplement (Figure S12), but refrain from any further analysis of the albedo DuERF due to uncertainty related to distinguishing the signal from the noise. Maps of the forcing for each of the terms of the Ghan (2013) decomposition are provided in the supplement Figures S10–S12.

Figure 3 highlights several key findings across models. MPI-ESM-1-2-HAM and EC-Earth3-AerChem exhibit the largest reductions in LWP; this aligns with their significant positive SW cloud DuERF. Conversely, NorESM2-LM is unique in demonstrating a substantial increase in IWP, consistent with its large positive LW cloud DuERF. In general, dust has a limited impact on the global mean cloud fraction. Models without aerosol-aware INP representations typically show a slight reduction in cloud fraction, particularly at low and mid-levels. In contrast, NorESM2-LM stands out by showing an increase in overall cloud fraction, mainly attributed to high clouds. With respect to Nd, the models generally agree on a slight reduction. In particular, EC-Earth3-AerChem records the largest decrease in Nd, over 3% relative to *piClim-control*. Dust can affect Nd through semidirect effects and by acting as a condensation sink for other aerosol tracers. The most consistent finding in Figure 3 is the change in precipitation. Eight of the nine models show a decrease in precipitation. In the following section, we examine the relationship between DuERF and precipitation change.

## 4  Relationship between dust forcing and precipitation change

Possibly the most notable result of Figure 3 is the large agreement between the models on the impact of dust to decrease precipitation. There are several different mechanisms that would lead to a reduction in precipitation in the models, such as decreased evaporation, increased stability, and changes in heating rates. Among the models with the largest decrease in precipitation, we

have NorESM2-LM (dust INPs, but highly scattering dust), GISS-E2-1-G, GFDL-ESM4 and IPSL-CM6A-LR-INCA (no dust INPs, but strongly absorbing dust).

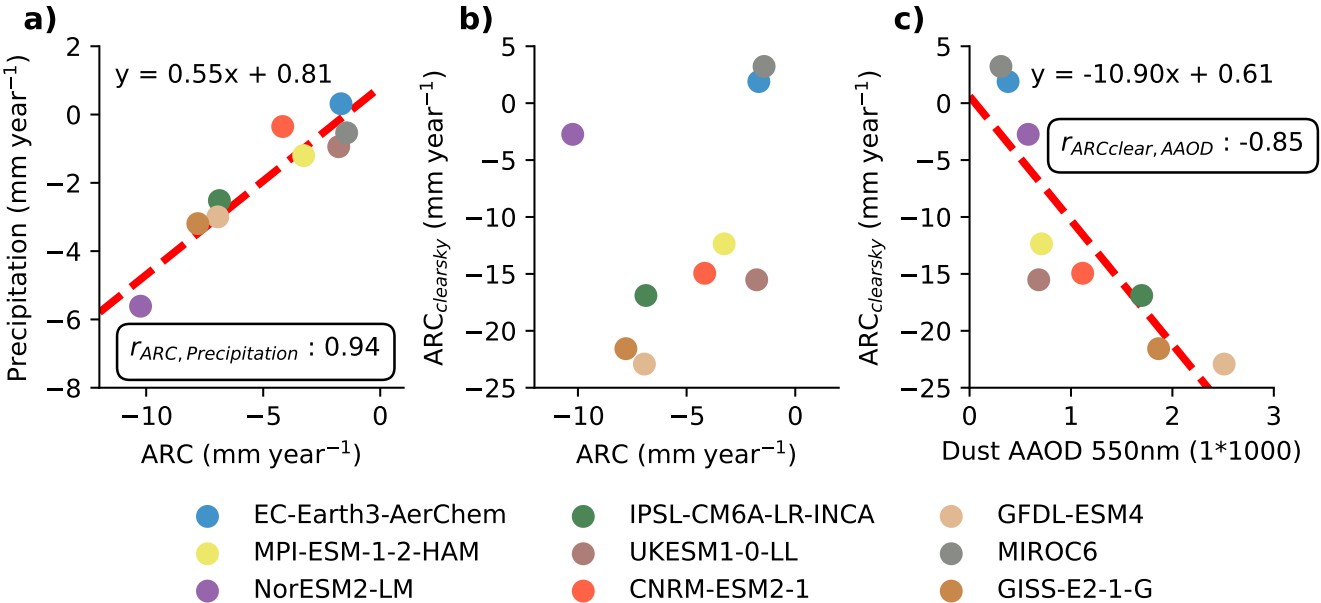

**Figure 4. a)** Change in Atmospheric radiative cooling (ARC) (mm year$^{-1}$) against precipitation change (mm year$^{-1}$) between piClim-control and piClim-2xdust. **b)** ARC against clear-sky ARC. **c)** Dust absorption (Dust AAOD) against clear-sky ARC. In panels **a)** and **c)**, the correlation coefficient r is displayed within rounded text boxes.

To understand dust-induced precipitation changes and the impact of dust INPs versus dust absorption, we analyse how dust perturbations affect ARC and how varying ARC contributes to inter-model differences in simulated dust-precipitation responses. The ARC is affected by changes in SW absorption, LW cooling of the atmosphere, and sensible heat fluxes at the surface. The clear sky changes in ARC, that is, in the absence of clouds, are primarily influenced by aerosol absorption. In Figure 4 we have converted the ARC into equivalent precipitation units (for details, see Supplement Section 1.1). Figure 4a shows how models with weakly absorbing dust, such as MIROC6 and EC-Earth3-AerChem, show no significant change in ARC or precipitation for both clear and all-sky conditions. NorESM2-LM exhibits notably weaker reduction clear sky ARC compared all-sky ARC. Models containing more absorbing dust display the opposite of NorESM2-LM by having substantially more clear sky heating compared to all-sky heating. Correlating the change in AAOD with clear sky ARC, reveals that, in models such as GISS-E2-1-G, IPSL-CM6A-LR-INCA, and GFDL-ESM4, dust absorption is the predominant cause of clear sky heating and precipitation inhibition. NorESM2-LM lacks significant dust absorption and therefore shows minimal change in clear-sky ARC. Rather, for NorESM2-LM, the precipitation decrease is driven by cloudy-sky ARC, related to increased high-altitude ice clouds that retain more of the outgoing LW radiation, warming the atmosphere, and lowering precipitation.

The effect of dust absorption on ARC operates largely independently of the LW effect from increased ice clouds, suggesting that these two effects—ice cloud changes in NorESM2-LM and SW absorption in others—need to be combined, to assess the maximum impact dust could have on precipitation in models. Consequently, we assess that doubling the dust load could decrease precipitation by up to approximately 10 mm year$^{-1}$. This magnitude is comparable to the inhibition of precipitation caused by adding anthropogenic black carbon (15 mm year$^{-1}$) (Samset, 2022) to the atmosphere. It is worth mentioning that the impact of dust on cirrus clouds and dust absorption exhibit different regional precipitation changes, as also shown by Zhao et al. (2024).

As an example from the AerChemMIP ensemble, we observe a distinct relationship between the ESMs that exhibit a relatively large MAC and thus produce a comparatively large increase in dust absorption over North Africa and increase in precipitation locally (see Supplement Figures S7 and S13). This indicates the role of dust absorption in determining the position of the Intertropical Convergence Zone (e.g., Pausata et al., 2016; Wilcox et al., 2010). Note that since the SSTs are fixed in the *piClim* experiments, the full response of the dust-perturbed climate system is not fully visible. For example, there is minimal dust cooling over the oceans because of the reduced SW radiation at the surface. Such cooling would lead to less evaporation and likely reduced precipitation in a fully coupled model setup (the slow precipitation response).

## 5    Conclusions

Dust is well established as an important factor in the Earth system owing to its diverse radiative impacts. The present study sheds light on how the CMIP6 generation of ESMs represents dust radiative effects and shows that model differences in dust representation have a major influence on the uncertainties in the DuERF. We decompose the DuERF into a contribution from dust-radiation interactions (direct DuERF) and dust-cloud interactions (cloud DuERF), which we further associate with intensive and extensive parameters that are influential for the DuERF in the models. We upped the number of models included from six as in Thornhill et al. (2021) to nine.

The simulated direct DuERF ranged from $-0.56$ to $+0.05$ Wm$^{-2}$. The inter-model spread in the SW direct DuERF forcing efficiency per DOD is largely consistent with the model differences in the dust MAC. The ESMs still have a large span in the MAC, which is tightly bound to the dust complex refractive index assumed in each model. This variability in MAC is similar to that previously reported (e.g., Gliß et al., 2021; Huneeus et al., 2011), because the models have not changed. Altogether, the variability in DOD and DAOD explains a large part (90%) of the spread in total and SW direct DuERF. The models show the most variation with respect to the TOA direct DuERF over the deserts, exposing that the planetary albedo calculated from the airborne dust in the models might not be internally consistent with the albedo of the desert surface. This inconsistency is showing up and is particularly revealing in some models that have strong TOA cooling or TOA warming over the desert.

Differences in the model size distribution of dust particles are an important cause of spread in simulated LW direct DuERF. Despite several models claiming that they use a more realistic size distribution at the point of emissions following brittle fragmentation theory (BFT) (Kok, 2011), the large variability in dust load (larger than dust AOD) indicates a high variability in coarse dust load between ESMs. Models that include a greater fraction of coarse to super-coarse dust can exhibit a LW

forcing efficiency that is orders of magnitude larger than models that under-represent the amount of coarse and super-coarse dust (Figure 2b). The underrepresentation of coarse dust has been shown to overestimate the negative values of direct DuERF by up to a factor of two (Kok et al., 2017). Furthermore, even ESMs that include dust size distribution that is more aligned with observational constraints would probably still underestimate LW direct DuERF due to neglecting LW scattering, which was only included in one of the nine AerChemMIP ESMs. Including LW scattering could increase direct LW DuERF by 20%–60% (Dufresne et al., 2002).

To allow for a more comprehensive assessment of the LW dust radiative effect in the future, ESMs should include diagnostics of AOD and AAOD at 10 $\mu$m. These diagnostics could also facilitate future multi-model evaluations against infrared emission measured from satellites (e.g., by the Infrared Atmospheric Sounding Interferometer (IASI), retrieving dust optical depth at 10 $\mu$m). Another approach would be to evaluate the dust size distribution in the models with observations. Formenti and Di Biagio (2024) compiled a comprehensive collection of in situ dust particle size measurements into a consistent dataset describing the dust particle size distribution and its evolution from emissions to deposition. By also providing a constraint on the evolution of the size distribution during transport, it offers an additional challenge for models to correct the size distribution not only at emissions, but also throughout its lifecycle. Accordingly, there are observational constraints available that can be used to significantly reduce the inter-model diversity in the direct DuERF.

The simulated cloud DuERF between the models ranges from $-0.04$ to $0.16\,\mathrm{W m^{-2}}$, this span is a conservative estimate, given that most of the AerChemMIP ESMs lack an aerosol-aware INP representation. NorESM2-LM, which includes an aerosol aware INP representation, exhibits the most substantial dust LW and SW cloud DuERF, showing an increase in cirrus cloud cover. However, the LW and SW radiative effects largely cancel each other out in NorESM2-LM, and we cannot conclude whether this would also be the case in other models. Besides NorESM2-LM, the other models exhibit a cloud DuERF mainly driven by dust semi-direct effects driven by dust absorption or dust affecting the CCN concentration, resulting in LW and SW cloud DuERF that are a factor of 2–3 lower than in NorESM2-LM.

The ESMs agree that atmospheric dust leads to a decrease in precipitation globally and is to the first order dependent on the amount of dust. However, the mechanisms driving the precipitation decrease differ. In NorESM2-LM increases in atmospheric absorption due to more cirrus clouds are largely responsible for the weaker ARC and the corresponding precipitation decrease. In the other models, dust SW absorption is the main contributor to precipitation inhibition. Together, the simulated reduction caused by dust absorption and the increase in cirrus clouds is comparable to the estimated precipitation inhibition due to anthropogenic black carbon. While globally atmospheric absorption leads to reduced precipitation, this is not necessarily the case for a given region. Changes in precipitation in North Africa correlate positively with the DuERF over the region (see Supplement Figure S8 and S13), indicating that warming over the Sahara invokes not only a change in ARC (hence precipitation) but also involves a change in the circulation, e.g., a shift in the ITCZ position.

A general conclusion from our analysis of the *piClim-2xdust* experiment, which is less apparent from the Thornhill et al. (2021) analysis, is that the dust emission strength is certainly just one of several factors that influence the DuERF. Among these factors are very likely the MAC, dust ice cloud interactions, dust size distributions, surface albedo vs. dust SSA, and LW absorption and scattering. The indirect effects of dust on $SO_2$/$HNO_3$ and secondary aerosol distributions are likely less

important in the pre-industrial simulations studied here, but could be important in an anthropogenically influenced climate (Klingmüller et al., 2019). In fact, several of the factors related to the dust representation that we are discussing lead to models that exhibit forcing efficiencies that can differ by a factor of ten between the models. To better sample the uncertainty in dust

forcing efficiency we would need more information on the whole parameter space that influences it in the models. Using a PPE would be a systematic approach in which multiple model parameters are varied simultaneously to most efficiently gather information about the parameter space of a given model (Sexton et al., 2021) affecting its DuERF. Then, using the PPE data to train an emulator of the full dust climate response of the ESM, which can then be used to rapidly generate model predictions. This can be an important way to explore the value of different observational constraints (Watson-Parris et al., 2021). Exposing

a larger set of models to a consistent set of observational constraints could be a game changer for reducing the inter-model differences in DuERF.

Our results have shown multiple differences in how the CMIP6 ESMs represent dust. These differences were shown to have a substantial impact on important aspects of the climate system, such as global precipitation and energy balance. With the growing number of studies providing evidence of drastic increases in the amount of dust worldwide in the last 150 years,

dust changes could have serious implications for how we understand the forcing history. Our results reinforce the point that dust-cloud interactions are more complex than the direct effect of dust and that their contribution to the DuERF should not be neglected. Additionally, this paper highlights the importance of discussing both SW and LW dust indirect effects. More focused attention to several key aspects of dust and climate interactions, particularly with regard to the representation of emissions, optical properties, and dust-cloud interactions is needed. Collaborative efforts across disciplines are critical to addressing these

challenges and improving the accuracy of dust modelling in the next generation of ESMs.

*Code and data availability.* Model output from AerChemMIP experiments used for creating figures are similar to that from Thornhill et al. (2021) with the exception of two model datasets added afterwards to the CMIP6 archive (MPI-ESM-1-2-HAM and EC-Earth3-AerChem). We thank the World Climate Research Programme and the Earth System Grid Federation for open access to the data of AerChemMIP (https://aims2.llnl.gov/search/cmip6/?mip_era=CMIP6&activity_id=AerChemMIP,%20WCRP,%202024a). Due to the nature of this analy-

595 sis, there is no model code associated with this article. The code used to create the figures shown in the manuscript is available on Zenodo (Haugvaldstad, 2025a, b).

*Author contributions.* OWH: Wrote the manuscript and did the analysis. MS: Provided supervision and gave feedback and comments to drafts of the manuscript. DO and TS: Gave feedback throughout the writing of the manuscript and helped revising drafts of the manuscript.

*Competing interests.* The authors declare no competing interest

*Acknowledgements.* We would like to acknowledge in particular the contribution from the modellers which participated in AerChemMIP and made the AerChemMIP multi-model dataset openly available on the CMIP6 database via the ESGF nodes. Without their efforts our analysis would not have been possible. Specifically we would acknowledge the following modellers: Martine Michou, Pierre Nabat and Roland Séférian (CNRM-ESM2-1), Twan van Noije (EC-Earth3-AerChem), David Neubauer (MPI-ESM-1-2-HAM), Dirk Olivié and Ada Gjermundsen (NorESM2-LM), Olivier Boucher, Yves Balkanski and Ramiro Checa-Garcia (IPSL-CM6A-LR-INCA), Fiona O'Connor, Gerd Folberth and Jane Mulcahy (UKESM1-0-LL), Larry Horowitz, Vaishali Naik and Fabien Paulot (GFDL-ESM4), Toshihiko Takemura (MIROC6) and Susanne Bauer and Kostas Tsigaridis (GISS-ModelE). We also acknowledge Casey Wall (University of Stockholm) for the insightful suggestion to examine the relationship between dust precipitation inhibition and the change in ARC.

The storage and computational resources for the analysis were provided by Sigma2—the National Infrastructure for High-Performance Computing and Data Storage in Norway.

We express our gratitude to the two anonymous reviewers for their constructive comments and suggestions to improve this manuscript.

*Financial support.* OWH is financed by the Norwegian Meteorological Institute's own funding. The Research Council of Norway funded parts of this work under grants 270061 (INES) and 295046 (KeyCLIM). This work has also received funding from the European Union's Horizon 2020 research and innovation programme under grant agreement No 821205 (FORCeS).

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

**Table 1.** Models used in this study including additional relevant information. *Lat./long.*: horizontal grid resolution. *Vert. Levs.*: number of vertical levels. *Ref. Emission scheme*: Reference for each ESMs' dust emission scheme. *Size char.*: Characteristics of the particle size distribution. *Dust refrac. index*: Refractive indices of dust at 550nm (real + imaginary part). *LW Scatt.*: LW scattering by dust represented? *CCN*: Does dust act as CCN? *INP*: Is dust represented as an INP? *Ghan*: Diagnostics required for Ghan (2013) decomposition? (Y/N). *Aerosol Scheme*: Name of aerosol module. *Model Ref.*: key references regarding the host model.

| Model | Lat°/long° | Vert. Levs. | Ref. Emission scheme | Size char. | Dust refrac. index | LW Scatt. | CCN (Y/N) | IN (Y/N) | Ghan (Y/N) | Aerosol Scheme | Model Ref. |
|---|---|---|---|---|---|---|---|---|---|---|---|
| EC-Earth3-AerChem | 2 × 3 (0.8 × 0.8) | 34 (91) | Tegen et al. (2002); Heinold et al. (2007) | (****) 0.05–0.5 (1.59), >0.5μm (2.0) | 1.52 + 0.0011i | N | Y(†) | N | Y | TM5 (M7) | van Noije et al. (2021) |
| MPI-ESM-1-2-HAM | 1.9 × 1.9 | 47 | Tegen et al. (2019) | (****) 0.05–0.5 (1.59), >0.5μm (2.0) | 1.52 + 0.0011i | N | Y(†) | Y | Y | HAM (M7) | Tegen et al. (2019) |
| NorESM2-LM | 1.9 × 1.9 | 32 | Zender et al. (2003) | (*) 0.22 (1.59), 0.63μm(2.0) | 1.53 + 0.0024i | N | Y | Y | Y | Oslo_Aero | Kirkevåg et al. (2013, 2018); Seland et al. (2020) |
| IPSL–CM6A-LR-INCA | 1.25 × 2.5 | 79 | Schulz et al. (1998) | (***) 2.50 (2.0) μm | 1.52 + 0.00147i | N | N | N | Y | INCA v 6.1 | Lurton et al. (2020); Balkanski et al. (2007); Hauglustaine et al. (2014) |
| UKESM1-0-LL | 1.25 × 1.88 | 85 | Woodward et al. (2022) | (**) 0.06324, 0.2, 0.6324, 2.0, 6.324, 20.0 63.24μm | 1.53 + 0.00148 | Y | N | N | Y | CLASSIC (dust) GLOMAP | Williams et al. (2018); Mulcahy et al. (2020) |
| CNRM-ESM2-1 | 1.4 × 1.4 | 91 | Séférian et al. (2019) | (**) 0.01, 1.0, 2.5, 20μm | 1.52 + 0.008i | N | N | N | Y | TACTIC_v2 | Séférian et al. (2019); Nabat et al. (2020) |
| GFDL-ESM4 | 1.0 × 1.2 | 49 | Ginoux et al. (2001) | (**) 1 (0.05), 2 (0.15) 3 (0.30), 6 (0.27), 10μm (0.23) | 1.49 + 0.00203i | N | N | N | N | Sectional | Zhao et al. (2018); Naik et al. (2013) |
| MIROC6 | 1.4 × 1.4 | 81 | Tegen et al. (2002); Takemura et al. (2009) | (**) 0.22 (0.0045), 0.46 (0.029), 1 (0.1766), 2.15 (0.2633), 4.64 (0.2633) 10.0 (0.2633) | 1.53 + 0.002i | N | Y | Y | N | SPRINTARS | Tatebe et al. (2019) |
| GISS-E2-1-G | 2 × 2.5 | 40 | Miller et al. (2006) | (**) 0.2, 0.5, 1, 2, 4, 8μm | 1.564 + 0.002i | N | N | N | N | OMA | Bauer et al. (2020) |

*Mode number mean radius, in parenthesis standard deviation . **Upper diameter of each size bin, in parenthesis (if available) emitted mass fraction in each bin. ***Mass median diameter, in parenthesis standard deviation. ****Dry radius interval, in parenthesis standard deviation. †Can be transferred from insoluble to soluble mode via heterogenous chemistry.