# Peer review of "Dust radiative forcing in CMIP6 Earth System models: insights from the AerChemMIP piClim-2xdust experiment"

_EGUsphere, 2025_

## Referee Comment (RC1)

This paper examines dust effective radiative forcing in nine CMIP6 Earth System Models using AerChemMIP experiments that double dust emissions under pre-industrial conditions. The total forcing was decomposed into direct and cloud (indirect) forcings. The intermodal differences were related to dust and cloud properties. The dust effective radiative forcing, especially the indirect forcing, is highly uncertain. Therefore, this is a meaningful work that helps to understand this uncertainty, and it is well within the scope of ACP. However, I find that some of the results are not supported by enough evidence and/or not fully explained/discussed. Therefore, I would recommend a major revision for this manuscript. Please find my general and specific comments below.

**General comments:**

1. About the dust impact on cirrus clouds in NorESM2-LM (L394-400). I think it is not plausible enough to say that the IWP and high cloud fraction increase are caused by cirrus cloud increase. First, high cloud may include some mixed-phase clouds, and IWP is a column integrated variable. In addition, the authors have not shown evidence supporting the statement that "in regions where heterogeneous ice nucleation predominates, additional INPs typically increase ice crystal concentrations, which appears to characterize NorESM2-LM". I suggest the authors to examine 3D zonal average variables (e.g., heterogeneous and homogeneous ice nucleation rate, ice water content, cloud fraction, ice number concentrations, temperature, etc.) to better support this statement. If this is not possible due to lacking output, I suggest the authors provide some discussion based on possible previous studies.

2. About $N_d$ decrease in EC-Earth-AerChem and MPI-ESM-1-2-HAM (L415-417) and other models. It is not clear to me why the increase in dust results in $N_d$ decrease. If dust does not activate as CCN, please specify (this does not seem to be the case for MPI-ESM-1-2-HAM as written in L155-156?). Also, how do the two models treat condensation of other aerosols/gases on dust? Do they assume that no secondary aerosol that can be CCN is formed on dust so that the condensation is a pure sink for, e.g., $SO_2$? In addition, the CCN decrease (Figure S6) is not necessarily related to "reducing formation of secondary aerosols". The authors may consider directly show changes in secondary aerosols and their precursor gases (e.g., sulfate, nitrate, $SO_2$, etc).

3. In section 3.1, the authors explain the variations in total DuERF mainly through dust properties related to direct forcing. How would indirect forcing and cloud properties impact DuERF? Also, in these models, DuERF is dominated by direct forcing, and the direct forcing is more uncertain than indirect forcing. This is different from the estimate by IPCC AR6, which shows that aerosol indirect forcing is larger and more uncertain. Please note this and add some discussion about this issue in the manuscript.

4. If possible, the authors may consider add the surface albedo forcing (Ghan, 2013) for completeness of the decomposition.

5. Some statements in the manuscript do not have proper citations. Please see my specific comments for details.

**Specific comments:**

L82-83: Please give references (i.e., which models do not include dust as CCN). Also please note that many ESMs actually do include dust as CCN, e.g., CESM and E3SM.

L88-89: "Dust readily … at warmer temperatures". Please give references about that dust can work as INPs and K-feldspar is more efficient.

L90-93: Please give references for the impact of dust on cirrus clouds.

L108 and the following lines: it is not clear to me how DuERF is determined when first reading this paragraph. It is better to clarify here that piClim-2xdust is compared to piClim-control. Also, it may be necessary to briefly mention that dust is the only perturbing factor, and all the others were kept the same as piClim-control.

L115: Do Thornhill et al. (2021) only examine total DuERF and not separate direct and cloud DuERF? If so, please specify.

L170-174: It is not clear to me whether dust impact heterogeneous ice nucleation in mixed-phase clouds in NorESM2. Does the bug results in no heterogeneous ice nucleation in mixed-phase clouds?

L207-208: Do you mean once other aerosols/precursor gases condense on dust, the hygroscopicity of dust does not change and no secondary aerosols are formed on dust (so dust is a pure sink for, e.g., sulfate and nitrate)? Please give references for this treatment.

Eq (5)-(7): I think the original equations in Ghan (2013) (see their Section 3) is clearer and more widely used.

L279-281: please add the estimate from Thornhill et al. (2021) here for comparison.

L400-401: This seems to be different from what being said in Section 2.2. Does the bug fully deactivate heterogeneous ice nucleation in the model?

L457-458: how did you make the assessment? Also, please give the exact number for BC inhabitation.

L462-463: I would say the models do not agree on the change in precipitation over North Africa, because four of them show decrease. Also, please show spatial distribution of precipitation change to support that ITCZ has shifted.

L478-480: This is not consistent with previous statement. It was claimed that surface albedo is relatively consistent among models in L269-271.

Figure 4: please check the unit for ARC. Also, does c) show dust AAOD in absolute value or its change?

Table 1: (1) Consider add a column showing if dust can be CCN or not. (2) Column MB95: does X mean N (no)? (3) Column Size char.: what are the numbers in parenthesis? (4) Please give reference for Ghan. (5) Please verify if GFDL-ESM4 has Ghan method or not.

---

## Author Comment (AC1)

**Comment:** This paper examines dust effective radiative forcing in nine CMIP6 Earth System Models using AerChemMIP experiments that double dust emissions under pre-industrial conditions. The total forcing was decomposed into direct and cloud (indirect) forcings. The intermodal differences were related to dust and cloud properties. The dust effective radiative forcing, especially the indirect forcing, is highly uncertain. Therefore, this is a meaningful work that helps to understand this uncertainty, and it is well within the scope of ACP. However, I find that some of the results are not supported by enough evidence and/or not fully explained/discussed. Therefore, I would recommend a major revision for this manuscript. Please find my general and specific comments below.

*Reply: We thank the reviewer for the insightful comments and suggestions. The reviewer's comments have meaningfully contributed to elevate the quality of the manuscript. We have added evidence and explanations where it was suggested. Line numbers refer to the tracked changes version of the revised manuscript. Blue text represents added new text in the manuscript.*

**General comments**

**Comment:** 1. About the dust impact on cirrus clouds in NorESM2-LM (L394-400). I think it is not plausible enough to say that the IWP and high cloud fraction increase are caused by cirrus cloud increase. First, high cloud may include some mixed-phase clouds, and IWP is a column integrated variable. In addition, the authors have not shown evidence supporting the statement that "in regions where heterogeneous ice nucleation predominates, additional INPs typically increase ice crystal concentrations, which appears to characterize NorESM2-LM". I suggest the authors to examine 3D zonal average variables (e.g., heterogeneous and homogeneous ice nucleation rate, ice water content, cloud fraction, ice number concentrations, temperature, etc.) to better support this statement. If this is not possible due to lacking output, I suggest the authors provide some discussion based on possible previous studies.

*Reply: We thank the reviewer for this question. The reviewer is of course right that the IWP is a column integrated quantity and therefore would include changes in cloud ice from both the cirrus and mixed-phase clouds. However, a bug in NorESM2-LM has the effect of disabling heterogeneous nucleation outside of the cirrus regime. The issue stems from a limiter in NorESM2 (CAM6-Nor) that was designed to prevent the tendencies of in-cloud ice number concentration from exceeding the maximum of what is possible based on the calculated number concentration of available ice nucleating particles (INPs). The problem with this limiter is that the tendencies for ice particle formation (immersion, contact, and deposition freezing) from the classical nucleation theory scheme, as described in Hoose et al. (2010), did not contribute to this limit. Therefore the Hoose scheme could not add to the ice number. This did not affect the parameterisation for heterogenous ice nucleation within cirrus clouds, which is activated based on a temperature threshold (below -37 C). Therefore, we are confident that the change in IWP is mainly driven by a cirrus cloud response. Figure S8, now added to the supplementary material, also confirms that the IWP response in the model is driven by changes in the cirrus cloud regime.*

[Figure]

**Figure S8:** *Zonal plot of NorESM2-LM piClim-control temperature, change in temperature between piClim-control and piClim-2xdust and change in ice mass fraction (kg/kg).The black solid line highlights the −37°C isotherm, which indicates the transition between the mixed-phase and cirrus cloud regimes.*

**Comment:** 2. About Nd decrease in EC-Earth3-AerChem and MPI-ESM-1-2-HAM (L415-417) and other models. It is not clear to me why the increase in dust results in Nd decrease. If dust does not activate as CCN, please specify (this does not seem to be the case for MPI-ESM-1-2-HAM as written in L155-156?). Also, how do the two models treat condensation of other aerosols/gases on dust? Do they assume that no secondary aerosol that can be CCN is formed on dust so that the condensation is a pure sink for, e.g., $SO_2$? In addition, the CCN decrease (Figure S6) is not necessarily related to "reducing formation of secondary aerosols". The authors may consider directly show changes in secondary aerosols and their precursor gases (e.g., sulfate, nitrate, $SO_2$, etc).

**Reply:** *We appreciate the reviewer's questions. We have now specified in Table 1 which models activate dust as CCN. The (small) decrease in Nd observed in EC-Earth3-AerChem and MPI-ESM1-2-HAM is consistent with several contributing factors, which we cannot disentangle based on the limited diagnostics available in CMIP6. Only those dust particles which get coated by soluble material become CCN. Note that the coating process is not necessarily increasing the number of CCN; the presence of more dust reduces likely the number of (fine) aerosol particles which can act as CCN, due to condensation of precursor gases and coagulation processes. Furthermore, the overall reduction in cloud fraction might also accompany a decrease in global mean Nd.*

**Q1:** Also, how do the two models treat condensation of other aerosols/gases on dust? Do they assume that no secondary aerosol that can be CCN is formed on dust so that the condensation is a pure sink for, e.g., $SO_2$?

**Reply:** *We have updated the description of how EC-Earth3-AerChem and MPI-ESM1-2-HAM treat condensation of aerosols on dust. In short, they represent coagulation of soluble modes on the dust particles and condensation of e.g. $H_2SO_4$, and these processes in the model can act to transfer the dust from the insoluble to soluble state. Dust in the*

*soluble state is considered a CCN. Therefore, dust is not merely a sink for aerosol precursor gases; if sufficient mass condenses onto a dust particle, it will be transferred to the soluble mode and be counted as a CCN.*

**Q2:** In addition, the CCN decrease (Figure S6) is not necessarily related to "reducing formation of secondary aerosols". The authors may consider directly show changes in secondary aerosols and their precursor gases (e.g., sulfate, nitrate, SO 2 , etc).

*Reply: Between the piClim-control and piClim-2xdust simulations, emissions of precursor gases remain unchanged. However, the increased dust does provide more surface area for condensation, resulting in a larger condensation sink for precursor gases, which subsequently reduces the new particle formation rate from these gases. Coagulation with dust can only decrease the number of cloud condensation nuclei (CCN), as soluble particles stick to initially insoluble dust particles, creating larger soluble particles. Furthermore, doubling the dust concentration would lead to a substantial reduction in radiation at the surface locally, resulting in less evaporation and cloudiness, which could also contribute to a decrease in Nd. Unfortunately, the diagnostics for changes in secondary aerosols and their precursor gases are quite limited and incomplete in the CMIP6 experiments. Therefore, we consider addressing these changes, as suggested by the reviewer, to be beyond the scope of this paper. However, we agree that adding such diagnostics would be very useful in future experiments. Given that only two models provided CCN diagnostics, we can offer little more than our hypothesis regarding the situation. We have revised the text to clarify that this is our hypothesis and that further information on CCN changes is needed to draw definitive conclusions. Karydis et al. (2017) did several modelling experiments investigating the impact of dust on Nd and found dust to decrease Nd in polluted regions, while increasing Nd over the deserts, overall they found dust to decrease Nd by 11% globally .*

**Comment:** 3. In section 3.1, the authors explain the variations in total DuERF mainly through dust properties related to direct forcing. How would indirect forcing and cloud properties impact DuERF? Also, in these models, DuERF is dominated by direct forcing, and the direct forcing is more uncertain than indirect forcing. This is different from the estimate by IPCC AR6, which shows that aerosol indirect forcing is larger and more uncertain. Please note this and add some discussion about this issue in the manuscript.

*Reply: We appreciate the reviewer's question and reply for the different questions separately:*

**Q1:** How would indirect forcing and cloud properties impact DuERF?

*Reply: The most recent Kok et al. (2023) assessment found that dust indirect effects would most likely have a slight positive effect. Therefore, if the ESMs made improvements on their representation of dust-cloud interactions we would expect the DuERF to be less negative, although uncertainties remain large.*

**Q2:** Also, in these models, DuERF is dominated by direct forcing, and the direct forcing is more uncertain than indirect forcing.

*Reply: The model spread is larger in terms of the direct DuERF than the cloud DuERF, however, this only tells us that the ESMs currently have larger diversity in how they represent dust direct radiative effects compared to indirect*

*radiative effects. Furthermore the majority of the ESMs do not represent many of the important processes that are necessary to represent the full range of dust indirect effects, e.g. missing aerosol-aware INP representation, thus interpreting the inter-model spread in cloud DuERF as the uncertainty in it would be inaccurate.*

**Q3:** This is different from the estimate by IPCC AR6, which shows that aerosol indirect forcing is larger and more uncertain.

*Reply: The indirect effect related to soluble aerosols (not primarily dust) is represented within the microphysics in more or less all climate models; therefore intermodel spread for total aerosol effects gives a more accurate depiction of the uncertainty, as stated in AR6. As we show in Figure 3a, individual LW and SW flux changes due to dust can be larger in magnitude than the dust direct radiative effects. This is unlike the indirect forcing by anthropogenic aerosols, that is mainly connected to changes in liquid clouds and droplet number concentration, which is well known to mainly have a SW cooling effect. In contrast, dust influences both liquid and ice clouds and the balance between SW and LW radiative effects can pull in either direction, making the overall dust cloud radiative effect weaker than that of anthropogenic aerosols. We acknowledge that this is still very uncertain.*

*We have added the following discussion to the manuscript see **line 570-580**:* Contrasting direct DuERF (Figure 2a) and cloud DuERF (Figure 3 a), we see that the inter-model spread and magnitude of DuERF are dominated by direct DuERF. However, the larger spread in direct DuERF should not be interpreted as the cloud DuERF being less uncertain compared to direct DuERF, as current ESMs cannot be trusted to accurately depict the uncertainty in dust-cloud interactions. This only shows that the ESMs currently have larger diversity in how they represent direct radiative effects of dust compared to indirect radiative effects. Given that most ESMs lack crucial processes for depicting dust-cloud radiative effects, e.g. aerosol-aware INP representation, the apparent model consistency is due to a lack of representation and not lack of uncertainty. The DuERF is also different from the anthropogenic aerosol ERF (e.g., IPCC_AR6 Forster et al., 2021), which shows that aerosol indirect forcing is the largest and most uncertain aspect of aerosol radiative forcing. However, the dust radiative effect is in several aspects different from the indirect effect of soluble aerosols; for example, dust influences both liquid and ice clouds, and the SW and LW radiative effects can pull in opposite directions (McGraw et al., 2020), making the overall dust cloud radiative effect appear weaker than that of anthropogenic aerosols.

**Comment:** 4. If possible, the authors may consider add the surface albedo forcing (Ghan, 2013) for completeness of the decomposition.

*Reply: We thank the reviewer for the suggestion and we have added maps of the surface albedo forcing to the supplement, together with maps of the Cloud DuERF, and Direct DuERF. See Figures S10 to S12. We also added some brief discussion around the Albedo DuERF, **See Lines 581-586**:*

The Ghan (2013) decomposition includes a `residual' term that is attributed to changes in albedo (Figure S12). With respect to the global mean value, the albedo DuERF ranges from $-0.01$ to $0.14$ Wm$^{-2}$, and except for NorESM2-LM and CNRM-ESM2-1, it is small below $0.05$ Wm$^{-2}$. The spatial distribution of the albedo forcing is also not consistent between the ESMs. Consequently, we provide the albedo term for completeness of the decomposition in

the supplement (Figure S12), but refrain from any further analysis of the albedo DuERF due to uncertainty related to distinguishing the signal from the noise. Maps of the forcing for each of the terms of the Ghan (2013) decomposition are provided in the supplement Figures S10 - S12.

**Comment:** 5. Some statements in the manuscript do not have proper citations. Please see my specific comments for details.

**Reply:** *We thank the reviewer for identifying these issues and we have addressed the specific comments below.*

**Specific comments**

**Comment: L88-89:** "Dust readily ... at warmer temperatures". Please give references about that dust can work as INPs and K-feldspar is more efficient.

**Reply:** *In the revised introduction this sentence was removed to focus the introduction more towards how ESMs represent dust radiative effects.*

**Comment:** *L90-93: Please give references for the impact of dust on cirrus clouds.*
**Reply:** *Added references to the impact of cirrus clouds.*

*Froyd, K. D., Yu, P., Schill, G. P., Brock, C. A., Kupc, A., Williamson, C. J., et al. (2022). Dominant role of mineral dust in cirrus cloud formation revealed by global-scale measurements. Nature Geoscience, 15(3), 177–183.* https://doi.org/10.1038/s41561-022-00901-w

**Comment:** L108 and the following lines: it is not clear to me how DuERF is determined when first reading this paragraph. It is better to clarify here that piClim-2xdust is compared to piClim-control. Also, it may be necessary to briefly mention that dust is the only perturbing factor, and all the others were kept the same as piClim-control.

**Reply:** *We thank the reviewer for pointing this out and we have updated the text to make this more clear.*
**Line 171-171.** We define DuERF as the difference in the TOA imbalance between *piClim-2xdust* and *piClim-control*, with the dust emission perturbation being the only factor that separates the two simulations.

**Comment:** L115 Do Thornhill et al. (2021) only examine total DuERF and not separate direct and cloud DuERF? If so, please specify.

**Reply:** *We thank the reviewer for pointing this out, indeed Thornhill et al. (2021) do only examine the net DuERF; we have updated the text accordingly.*
**Line 183-184**: This article expands on the results of Thornhill et al. (2021), by quantifying the direct and cloud DuERF in the models, which was not shown in the Thornhill paper.

**Comment:** L170-174: It is not clear to me whether dust impact heterogeneous ice nucleation in mixed-phase clouds in NorESM2. Does the bug results in no heterogeneous ice nucleation in mixed-phase clouds?
**Reply:** *Yes this bug negates the ability of the Hoose scheme to modify the cloud ice numbers. See our reply to comment 1 for details.*

**Comment:** L207-208: Do you mean once other aerosols/precursor gases condense on dust, the hygroscopicity of dust does not change and no secondary aerosols are formed on dust (so dust is a pure sink for, e.g., sulfate and nitrate)? Please give references for this treatment.

**Reply:** *Dust aerosols are represented as internally mixed in the OMA scheme, however, dust is not included in the hygroscopic mass fraction of aerosols that can participate in the cloud nucleation processes (Schmidt et al., 2014). We have added a reference and rephrased the sentence.*

**Line 287-289:** Dust aerosols do not directly impact cloud droplet concentration  because dust is not included in the hygroscopic mass fraction of aerosols that can participate in the cloud nucleation processes (Schmidt et al. 2014).

**Comment:** Eq (5)-(7): I think the original equations in Ghan (2013) (see their Section 3) is clearer and more widely used.

**Reply:** *We thank the reviewer for this comment. The original equation might indeed be better known. We chose to display the equations as written in the manuscript to make it clear, which CMIP6 CMOR variables were used. However, to accommodate readers that are familiar with the Ghan (2013) equation, we now show the original equation next to the equation containing the CMOR variable name.*

**Comment:** L279-281: please add the estimate from Thornhill et al. (2021) here for comparison.

**Reply:** *Thanks for pointing out this issue, we have added the range from Thornhill et al.*

**Line 382-383:**  from $0.09 \text{Wm}^{-2}$ to $-0.41 \text{Wm}^{-2}$ compared to $0.09 \text{Wm}^{-2}$ to $-0.18 \text{Wm}^{-2}$ reported in Thornhill et al. (2021).

**Comment:** L400-401: This seems to be different from what being said in Section 2.2. Does the bug fully deactivate heterogeneous ice nucleation in the model?

**Reply:** *We thank the reviewer for identifying this issue and we have revised the text accordingly.* **See line 529- 530**

**Comment:** L457-458: how did you make the assessment? Also, please give the exact number for BC inhabitation.

**Reply:** *We appreciate the reviewer's question. We made this assessment by considering that the two mechanisms are related in that both BC and dust lead to a warming of the atmosphere. The assessment obviously compares the dust (anth. BC) impact on precipitation to a reference case without dust (anth. BC) (clarified in the text). We have added the exact number for the BC precipitation from Samset (2022) in the revised manuscript.* **See Line 621-622**

*Samset, B. H. (2022). Aerosol absorption has an underappreciated role in historical precipitation change. Communications Earth & Environment, 3(1), 1–8. https://doi.org/10.1038/s43247-022-00576-6*

**Comment:** I would say the models do not agree on the change in precipitation over North Africa, because four of them show decrease. Also, please show spatial distribution of precipitation change to support that ITCZ has shifted.

*Reply:* *We thank the reviewer for this comment. We agree that the original statement is somewhat inaccurate, and we have now revised the text to indicate that the majority of the models exhibiting the largest increase in Dust Absorption Optical Depth (DAOD) over North Africa also show an increase in precipitation across the region. We have for completeness added a figure showing the spatial distribution of the precipitation change (see Figure S13). See line 689-692 for text changes.*

**Comment:** L478-480 This is not consistent with previous statement. It was claimed that surface albedo is relatively consistent among models in L269-271.

*Reply:* *We thank the reviewer for pointing out this issue. The way the statement was written was indeed ambiguous. What we were trying to say is that the models are not internally consistent on how they represent the albedo of the surface versus the albedo from airborne dust. Thus the models have a too large contrast between the surface and the dust above, a scene that in reality should not be very contrasting. We have rephrased the sentence in the revised manuscript. See Line 645-649*

**Comment:** Figure 4: please check the unit for ARC. Also, does c) show dust AAOD in absolute value or its change?

*Reply:* *We thank the reviewer for noticing this issue. The ARC is converted to equivalent precipitation units to make it easier to relate ARC back to precipitation change. This conversion is now explained in the supplement (see Section S1.1). The DAOD shown is the change between piClim-2xdust and piClim-control. We updated the figure caption accordingly.*

**Comment: Table 1:** (1) Consider adding a column showing if dust can be CCN or not. (2) Column MB95: does X mean N (no)? (3) Column Size char.: what are the numbers in parenthesis? (4) Please give reference for Ghan. (5) Please verify if GFDL-ESM4 has Ghan method or not.

*Reply:* *We appreciate the reviewers suggestions and have made the following changes to Table 1. We removed the MB95 column. Instead we added one column showing which ESMs include LW scattering and one column showing which models include dust as CCN. The GFDL-ESM4 model does have the Ghan diagnostic and we have updated Table 1 accordingly. We thank the reviewer for noticing this error. For the updated table please see the tracked changes version of the manuscript.*

---

## Author Comment (AC2)

The manuscript presents a comprehensive analysis of dust effective radiative forcing (DuERF) using CMIP6 ESMs under the AerChemMIP piClim-2xdust experiment, contrasted with the piClim-control run. The study decomposes DuERF into direct and indirect components, evaluates model uncertainties, and quantifies the impact of dust perturbations on precipitation. While the manuscript is well-structured and provides valuable insights, several key issues need addressing to improve clarity, interpretability, and robustness. Below, I outline major revisions the authors may want to consider to ensure the study meets its full potential.

*Reply: We thank the reviewer for the valuable comments and suggestions, this has really helped elevate the quality of the manuscript. Our reply to the reviewer comments are written in italic and line changes are referring to the tracked changes version of the revised manuscript. Blue text refers to added text.*

**General comments**

**Comment:** 1. The introduction is quite lengthy and detailed, and it gives me the impression that the authors aim to constrain the uncertainty in dust forcing estimates by addressing the factors that contribute to this uncertainty. However, this is not the primary goal of their study which I did not realize until the last several sentences of the introduction section. To better align with the study's goals the introduction could be streamlined by shortening the discussion of factors contributing to uncertainty and focusing more on how those factors are represented in ESMs and how they influence the DuERF.

*Reply: We thank the reviewer for their insightful feedback regarding the introduction. The introduction has been revised accordingly, please see the tracked changes version of the revised manuscript for the specific changes.*

**Comment:** 2. The study analyzes DuERF using the idealized piClim-2xdust experiment, which perturbs dust emissions under preindustrial conditions. While the emission increase (70-105%) is comparable to or larger than estimated historical changes (Kok et al., 2023), the experimental design, fixed SSTs and no anthropogenic forcing, means the results reflect model-specific dust responses rather than real-world historical forcing. The authors should clarify that these findings cannot be directly compared to studies quantifying DuERF over the industrial era, as the mechanisms and climate feedback differ. This distinction is critical to avoid misinterpretation of the DuERF values presented.

*Reply: We thank the reviewer for their valuable comment regarding the interpretation of our results in the context of the piClim-2xdust experiment. We have made changes to the manuscript accordingly, and we now explicitly state the distinctions between the DuERF derived from the piClim-2xdust experiment and model simulations targeted to quantify the real world DuERF.*

**Line 171 - 178:** We define DuERF as the difference in the TOA imbalance between *piClim-2xdust* and *piClim-control*, with the dust emission perturbation being the only factor that separates the two simulations.

Although the relative increase in dust in the *piClim-2xdust* is comparable in magnitude to the estimated real world historical change, it is important to note the distinction between DuERF and dust effects diagnosed from this idealised setting and real-world historical dust forcing. Specifically, sea surface temperatures (SSTs) are fixed, anthropogenic aerosols are set to preindustrial conditions, and the change in dust emission is imposed uniformly across dust source regions. Therefore, our findings cannot be directly compared with studies quantifying DuERF during the historical era (Leung et al., 2025). However, this idealised setting is still useful for investigating how ESMs behave in response to changes in dust burden.

**Comment:** 3. The authors need to explicitly state whether the ESMs account for LW aerosol scattering, if any, both in the model physics and in the reported LW and NET (SW+LW) forcing estimates. Omitting LW scattering likely biases the NET DuERF, as it can contribute 50% (20-60%) of the dust LW direct radiative effect at the TOA (Dufresne et al., 2002). If LW scattering is not included, please consider estimating the potential bias in the LW and NET DuERF and, if possible, adding this information to the LW and NET forcing estimates.

Also, while the manuscript notes weak LW direct forcing efficiencies (Figure 2b), it does not clarify whether this is due to missing physics (e.g., neglect of LW aerosol scattering, not only for coarse and super-coarse dust particles, although this is likely the major source), deficiencies in the size distribution (e.g., underrepresentation of super-coarse dust), or a combined effect of the two. I request clarification on these aspects.

**Reply:** *We thank the reviewer for highlighting these important points. All of the ESMs considered in our study account for the LW absorption of dust. However, only UKESM1-0-LL includes both LW absorption and scattering. The ESMs with a large mass extinction coefficient (MEC) are likely exhibiting deficiencies in the size distribution of dust particles. For instance, NorESM2-LM and EC-Earth3-AerChem have MEC values that align more closely with PM2.5 dust rather than PM20 (Kok et al., 2021), which indicates underestimation of coarse dust.*

*To address these issues, we have updated the table to include a column indicating whether each model represents LW scattering. We have also added a discussion as a part of the conclusion section of the text regarding the impact of missing LW scattering on DuERF.* **See line 650 - 665.**

 Differences in the model size distribution of dust particles are an important cause of spread in simulated LW direct DuERF. . Despite several models claiming that they use a more realistic size distribution  at the point of emissions following brittle fragmentation theory (BFT) (Kok, 2011), the large variability in dust  load (larger than  dust AOD) indicates a high variability in coarse dust  load between ESMs. Models that include a greater fraction of coarse to coarse-super dust can exhibit LW forcing efficiency that is orders of magnitude larger than models that underrepresent coarse and super-coarse dust. Furthermore,  even ESMs that

include dust size distribution more aligned with observational constraints would likely still underestimate LW direct DuERF due to neglecting LW scattering, which was only included in one of the nine AerChemMIP ESMs. Including LW scattering could increase direct LW DuERF by 20% to 60% (Dufresne et al., 2002).

**Comment:** 4. The outlier behavior of NorESM2-LM is attributed to a known bug (line 400) that largely disables heterogeneous ice nucleation in mixed-phase clouds (lines 171-172). However, it is unclear how the authors handle the indirect forcing results from this model, particularly in relation to the indirect forcing. Was this model excluded from the analysis or given reduced weight in the ensemble mean? The authors should either quantify the impact of this bug on the results or provide a clear justification for including the model and explain how its known bias was accounted for.

*Reply:* *We thank the reviewer for their observations regarding NorESM2-LM. This model was included in the multimodel mean for the indirect forcing estimates, and we weighted all models equally that provided the necessary diagnostics for assessing cloud DuERF.*

*NorESM2-LM was included in the multimodel mean for the indirect forcing estimates. We weighted all the models that provided the diagnostics needed for diagnosing the cloud DuERF equally. The bug in NorESM2-LM made the behavior of NorESM2-LM more similar to the models that did not have aerosol-aware INP representation for mixed-phase clouds. Therefore we do not judge NorESM2-LM to be any more unreliable than the models that do not represent the impact of dust INP at all. In contrast to most of the models, NorESM2-LM actually still takes dust effects on cirrus clouds into account, making it relatively advanced when it comes to dust influence on cold clouds despite the bug that disabled any dust influence on mixed-phase clouds.*

*To clarify, we have added a detailed explanation of this bug in the methods section (see also response to Reviewer 1, Comment 1).*

5. The manuscript provides a good analysis of dust-cloud interactions and precipitation responses. However, the model descriptions in Table 1, and, in fact, throughout the manuscript, lack critical details about how cloud microphysics and precipitation processes are represented in each ESM. Only limited information is scattered throughout the text. Given that these processes are central to the study's conclusions, particularly regarding dust indirect forcings and hydrological cycle impacts, a more comprehensive summary of model physics would significantly strengthen the interpretation of results.

*Reply:* *First and foremost, we do not examine the microphysics in the models, while acknowledging that the models have quite different microphysical schemes. Please note that the CMIP6 models and their schemes have been extensively documented in other sources. Our main point is that the precipitation response we see in the models is more of a thermodynamic response than dust acting to drastically alter precipitation through different microphysics. We use the ARC, which is a bulk thermodynamic quantity, and that has been shown to explain reductions in*

*precipitation (e.g., Pendergrass and Hartmann (2014)). We also point to the influence of absorption by dust to drive part of the inter-model differences in ARC. We acknowledge that dust also might impact ARC differently in the models due to differences in microphysics, but we deem further investigation of such impacts out of scope for this study, mainly because of missing diagnostics. We have added, however, some relevant microphysics information regarding representation of dust as INP and CCN to Table 1.*

*Pendergrass, A. G., & Hartmann, D. L. (2014). The Atmospheric Energy Constraint on Global-Mean Precipitation Change. https://doi.org/10.1175/JCLI-D-13-00163.1*

6. The manuscript would benefit from greater consistency in its use of aerosol optical metrics to characterize dust impacts. While the study appropriately focuses on dust radiative forcing, there appears to be an inconsistent application of optical depth metrics throughout the analysis. At times, the authors examine total aerosol optical depth (AOD) and absorption aerosol optical depth (AAOD), while at other points they reference dust optical depth (DOD). Unless I missed it, the authors did not provide dust absorption optical depth (DAOD). It is also possible that the authors are using AAOD to refer to what is defined here as DAOD. However, this should be clarified to avoid confusion.

This variation in metrics could potentially lead to confusion when interpreting results, as total AOD/AAOD incorporates contributions from all aerosol species, not just dust. Given that the study specifically investigates dust impacts through the piClim-2xdust experiment, the consistent use of dust-specific metrics (DOD and DAOD) would provide a more direct and unambiguous estimate of dust aerosol effects.

*Reply: We thank the reviewer for this insightful comment regarding the consistency of aerosol optical metrics in our manuscript. The model diagnostic that we use is the difference in AOD between piClim-2xdust and piClim-control, and since dust is the only factor (apart from minor aerosol responses in a dust-perturbed preindustrial atmosphere) impacting the change in AOD it does correspond to DOD. For the absorption AOD, note that DAOD is not a standard AerChemMIP diagnostic and thus the only way to get the dust absorption component is by taking the difference. We have updated the text according to the reviewer's suggestion. To avoid confusion with AOD, we have decided, as the reviewer suggested, to refer to the change in AOD and AAOD as DOD and DAOD throughout the manuscript.*

7. The references are not properly cited. I have pointed this out in a few places, but I encourage the authors to check for similar issues elsewhere in the manuscript.

I notice that the authors super frequently cite some frequencies, which are undoubtedly very valuable contributions to the research community. However, it is important to also acknowledge and appropriately credit the original studies that underpin key findings.

*Reply:* *We agree with the reviewer, it is important to value and acknowledge the original publications and have done our best to improve on this aspect the manuscript*

**Minor comments**

**Comment:** Line 35-36: What does this "environment changes" specifically refer to?

**Reply:** *This line has been removed in the revised introduction.*

**Comment:** Line 51-52: This statement is not strictly accurate. Several ESMs are capable of representing dust aerosols as distinct mineral components. A few examples are provided below, although the list is not exhaustive and continues to grow as model development advances.

This list here (and after) is provided for my convenience, as the items are more readily available to me than the others. The authors should feel entirely free to cite any of them or any other sources they find relevant:)

Gómez Maqueo Anaya, S., Althausen, D., Faust, M., Baars, H., Heinold, B., Hofer, J., Tegen, I., Ansmann, A., Engelmann, R., Skupin, A., Heese, B., and Schepanski, K.: The implementation of dust mineralogy in COSMO5.05-MUSCAT, Geosci. Model Dev., 17, 1271–1295, https://doi.org/10.5194/gmd-17-1271-2024, 2024.

Li, L., Mahowald, N. M., Miller, R. L., Pérez García-Pando, C., Klose, M., Hamilton, D. S., Gonçalves Ageitos, M., Ginoux, P., Balkanski, Y., Green, R. O., Kalashnikova, O., Kok, J. F., Obiso, V., Paynter, D., and Thompson, D. R.: Quantifying the range of the dust direct radiative effect due to source mineralogy uncertainty, Atmos. Chem. Phys., 21, 3973–4005, https://doi.org/10.5194/acp-21-3973-2021, 2021.

Gonçalves Ageitos, M., Obiso, V., Miller, R. L., Jorba, O., Klose, M., Dawson, M., Balkanski, Y., Perlwitz, J., Basart, S., Di Tomaso, E., Escribano, J., Macchia, F., Montané, G., Mahowald, N. M., Green, R. O., Thompson, D. R., and Pérez García-Pando, C.: Modeling dust mineralogical composition: sensitivity to soil mineralogy atlases and their expected climate impacts, Atmos. Chem. Phys., 23, 8623–8657, https://doi.org/10.5194/acp-23-8623-2023, 2023.

Li, L., Mahowald, N. M., Kok, J. F., Liu, X., Wu, M., Leung, D. M., Hamilton, D. S., Emmons, L. K., Huang, Y., Sexton, N., Meng, J., and Wan, J.: Importance of different parameterization changes for the updated dust cycle modeling in the Community Atmosphere Model (version 6.1), Geosci. Model Dev., 15, 8181–8219, https://doi.org/10.5194/gmd-15-8181-2022, 2022.

Perlwitz, J. P., Pérez García-Pando, C., and Miller, R. L.: Predicting the mineral composition of dust aerosols – Part 1: Representing key processes, Atmos. Chem. Phys., 15, 11593–11627, https://doi.org/10.5194/acp-15-11593-2015, 2015.

Scanza, R. A., Mahowald, N., Ghan, S., Zender, C. S., Kok, J. F., Liu, X., Zhang, Y., and Albani, S.: Modeling dust as component minerals in the Community Atmosphere Model: development of framework and impact on radiative forcing, Atmos. Chem. Phys., 15, 537–561, https://doi.org/10.5194/acp-15-537-2015, 2015.

**Reply**: *We thank the reviewer for the additional references. The reviewer is right that many models do have the ability to represent the distinct mineral components of dust, but we have yet to find any examples of models using this representation for their CMIP production simulations due to the large computational burden including additional tracers for each mineral component. However, the original sentence did not properly reflect these nuances, and we have now rewritten the text to better reflect those nuances.* ***See line 77-90***

**Comment:** Line 54: The current citation refers to observational data; however, the authors should instead cite modeling studies that have incorporated updated refractive indices consistent with these measurements. Two relevant studies are listed below. The first directly incorporates the measured refractive indices, while the second constrains iron oxide optics to ensure that the modeled dust optical characteristics are consistent with the measurements.

Wang, H., Liu, X., Wu, C., Lin, G., Dai, T., Goto, D., ... & Shi, G. (2024). Larger dust cooling effect estimated from regionally dependent refractive indices. Geophysical Research Letters, 51(9), e2023GL107647.

Li, L., Mahowald, N.M., Gonçalves Ageitos, M. et al. Improved constraints on hematite refractive index for estimating climatic effects of dust aerosols. Commun Earth Environ 5, 295 (2024). https://doi.org/10.1038/s43247-024-01441-4

**Reply:** *We thank the reviewer for the suggested references and have included them where appropriate.* **See line 85 -89**

**Comment:** Line 78-79: The term "pure dust" requires clarification. Do you mean particles composed solely of mineral components (though not limited to a single mineral species)? If so, this should be explicitly stated. Moreover, the claim that "pure dust is insoluble" is not strictly accurate. Some of the mineral components like hematite are indeed largely insoluble in water under ambient atmospheric conditions, but some others like calcite are highly soluble, and these are often present in atmospheric dust in small but non-negligible amounts.

**Reply:** *We agree that the term pure dust is not the most appropriate in the context of the study and have removed it. We agree that models assuming dust being "insoluble" is a simplification, ignoring the variable nature of dust particles. We have clarified in table 1 whether models have dust particles to act as CCN and further in the models description paragraphs which processes and assumptions led dust acting as CCN.*

**Comment:** Line 88-89: A reference is needed here, at minimum for K-feldspar. The following is one example that could be cited.

Atkinson, J. D., Murray, B. J., Woodhouse, M. T., Whale, T. F., Baustian, K. J., Carslaw, K. S., ... & Malkin, T. L. (2013). The importance of feldspar for ice nucleation by mineral dust in mixed-phase clouds. Nature, 498(7454), 355-358.

*Reply: In the revised manuscript this sentence has been removed.*

Line 103-105: The two sentences seem to contradict one another: "Furthermore, the net dust effective radiative forcing (DuERF) varies across models…" and "models may appear consistent in DuERF".
The first sentence states that DuERF varies between models, suggesting inconsistency, while the second says models "may appear consistent" which is the opposite.

*Reply: In the revised manuscript this sentence has been removed.*

**Comment:** Line 112: It would be helpful to specify which five models are being referred to.

*Reply: We appreciate the reviewers suggestion and have specified the models in the revised manuscript. **See Line 179-183***

Line 133-134: "wind dependent dust emission schemes". The current wording is somewhat informal, as dust emission in many ESMs depends on more than just wind speed. Additional factors such as the extent of bare soil, soil texture, and surface aridity also play critical roles in determining dust source regions.

*Reply: We agree with the reviewer's comment and have updated the text accordingly, see **Line 205- 206**.*

Line 150: Can the authors specify the composition of the mineral mixture used in this calculation?

*Reply: We thank the reviewer for the question. The internal mixture is between mineral dust and other aerosols: "When black carbon, dust, or both are present in the mix, these are treated as inclusions in a homogeneous background medium, using the Maxwell Garnett mixing rule." — von Noije et al. (2021). In other words, the statement does not refer to a mixture of minerals. EC-Earth3-AerChem uses a single set of refractive indices to represent mineral dust.*

**Comment:** Line 151: It would be useful to clarify how this model, along with others referenced, treats LW aerosol scattering.

*Reply: We thank the reviewer for the useful suggestions, we have included the treatment of LW scattering as a new column to Table 1.*

**Comment:** Line 191-192: In addition to the complex refractive index listed in Table 1, which specific optical properties are held fixed, e.g., single scattering albedo, extinction coefficient? Do these properties lack both spatial and temporal variability in the model?

*Reply: We thank the reviewer for pointing out ambiguities in the text. What we mean is that dust in the CNRM model is externally mixed and thus there is not any interaction between dust and other atmospheric species and accordingly the optical properties (SSA, MEC) of dust in each of the three bins do not change with time. In the revised manuscript we have clarified this.*

**Comment:** Line 269-271: This sentence should be revised for clarity or just removed. Since surface albedo is consistent across the models, it does not contribute to the spread in forcing efficiencies. This point is already clearly conveyed in the following sentence.

*Reply: We thank the reviewer for noticing this and we agree it was not a meaningful sentence, we have revised the sentence for clarity.* ***See line 360-366***

**Comment:** Line 313: Are there any insights into why these two models produce notably different results? Identifying specific parameterizations or assumptions that set them apart would strengthen the analysis.

*Reply: See reply to comment question 3. We now point to how UKESM1-0-LL is the only model representing LW scattering effects.*

**Comment:** Line 355: Why focus on AOD and AAOD rather than DOD and ADOD, which are more directly attributable to dust aerosols? Using dust-specific metrics would provide a clearer assessment of dust aerosol impacts, right?

*Reply: See our response to question 6.*

**Comment:** Line 362-363: The uncertainty range reported by Ridley et al. (2016) pertains to present-day DOD. It is unclear why the authors compare this with the modeled change in total AOD under preindustrial conditions.

*Reply: We appreciate the reviewer's question and hope this is partly clarified by our response to question 6. Since the dust emissions are the only factor changing, the difference in AOD between piClim-2xdust and piClim-control, representing a doubling of dust, should be a good approximation of the DOD in piClim-control. Furthermore the dust emissions in the models have not significantly changed between present day and pre-industrial. It is useful to compare the AOD perturbation (=preindustrial DOD) to that of the best available DOD of present day conditions. It also is useful to put in perspective the difference between piClim-2xdust and piCim-control comparing it to the present day dust effective radiative effect, especially for the direct effect, which changes rather linearly as a function of burden and optical depth.*

**Comment:** Line 379: I expect the authors to further propose, in addition to constraints on DOD, what other specific constraints should be included to reduce the uncertainty in the direct DuERF.

*Reply: We thank the reviewer for the suggestion and have added the following sentence **See line 505-508***

Going forward, we need to expose ESMs to a larger set of constraints on different aspects of the dust cycle, for example, particle size distribution (Kok et al., 2021), CRI (Li et al., 2024; Wang et al., 2024) or spatial gradients in DOD to constrain the lifetime of dust to reduce the uncertainty in the direct DuERF.

**Comment:** Line 480: Could the authors clarify the distinction between "dusty surface albedo" and the "planetary albedo effect from airborne dust"?

*Reply: We agree as written this was somewhat unambiguous, we have revised the text accordingly **See Line 645-649***

**Comment:** Line 517-518: Are there any references, figures, or tables provided to support this statement?

*Reply: We thank the reviewer for the question. We agree that the results we show in the manuscript do not offer sufficient evidence to support the claims of the statement as it was written, we have changed the wording of the statement and added a reference. **See Lines 697-698***

---

## Author Response (AR2)

**Minor Comments Reviewer 1**

The authors have done a nice and sufficient job addressing my previous comments. I have two very minor follow-up comments.

L621-620. It is still not clear to me how mathematically the precipitation decrease (10 mm/year) is estimated. Did you use any of the equations shown in Figure 4? The authors have mentioned that this is calculated by comparing to a reference case without dust at all. So, does it represent the dust effect on precipitation in your piClim-control cases (i.e., piClim-control – no dust)? Or does it represent the effect of doubling dust in piClim-2xdust comparing to piClim-control?

*Reply: We agree that as written the statement is confusing, we have updated the text to emphasize that a doubling of dust could cause a precipitation of up to 10mm/year. It is as the reviewer states, the precipitation decrease is estimated by comparing piClim-2xdust with piClim-control.*

 Consequently, we assess that doubling the dust load could decrease precipitation by up to approximately 10 mm year−1.

Please check all the references to supplemental figure numbers. There are some mistakes. For example, at L524 in the track changes version, I believe it should be Figure S9, instead of Figure S8. Also, at L626, it should be Figure S8.

*Reply: We thank the reviewer for identifying these errors and we have checked and corrected incorrect references to in the revised manuscript.*

**Minor Comments Reviewer 2**

I have only a few very minor comments on the revised manuscript, and I believe it could be accepted for publication thereafter.

Lines 38-40: To my knowledge, Marx et al. (2024) present results focused solely on the northwestern Pacific. I do not think their study can be used to support the statement regarding a global increase in dust burden.

*Reply: Yes the citation was inaccurate given the original statement. We have rephrased the sentence:*

Substantial evidence indicating that  atmospheric dust burden has  significantly increased in several regions around the globe since the beginning of the industrial era has  been established by observations (Hooper and Marx, 2018; Marx et al., 2024; Mulitza et al., 2010),

Lines 70-71: The citation of Claquin et al. (2003) might be questionable in this context, as their study investigates the effects of different mixing states of iron oxides (hematite and goethite) and, more generally, the influence of dust mineralogy on dust radiative forcing. Although variations in mineralogy can indeed alter the CRI of dust aerosols and, consequently, the DuERE, the connection between Claquin et al.'s findings and the specific claim in this sentence is somewhat indirect.

*Reply: We have removed the Claquin et al (2003) citation and added Myhre and Stordal, 2001; Li et al., 2021, modelling studies emphasizing the effect of CRI of dust on the magnitude of DuERE.*

Line 73: I am not entirely certain whether particle shape should be considered a dominant factor in determining the DuERE. However, based on the authors' citation of Ito et al. (2021), dust asphericity, under their modeling assumptions, leads to a relative change of approximately 15% in the top-of-atmosphere dust shortwave direct radiative effect (see their Table 5).

***Reply:*** *We have added one sentence describing the findings of Ito et al 2021. While we agree that particle size is not the dominant factor in the uncertainty of the DuERE at the TOA, the study shows that asphericity increases the surface cooling by almost 40%.*

Ito et al.(2021) found that dust asphericity alone increased the SW TOA cooling by around 15% (−0.32 vs. −0.28 W m−2 on a global scale), however, asphericity had limited impact on net TOA DuERE due to increased LW warming.

Lines 74-75: It may be helpful to include additional or alternative references, either from the list below or others known to the authors.

Laboratory studies:

Di Biagio C, Formenti P, Balkanski Y, Caponi L, Cazaunau M, Pangui E, Journet E, Nowak S, Andreae MO, Kandler K, Saeed T. Complex refractive indices and single-scattering albedo of global dust aerosols in the shortwave spectrum and relationship to size and iron content. Atmospheric Chemistry and Physics. 2019 Dec 19;19(24):15503-31.

Lafon S, Sokolik IN, Rajot JL, Caquineau S, Gaudichet A. Characterization of iron oxides in mineral dust aerosols: Implications for light absorption. Journal of Geophysical Research: Atmospheres. 2006 Nov 16;111(D21).

Modeling study:

Li, L., Mahowald, N. M., Miller, R. L., Pérez García-Pando, C., Klose, M., Hamilton, D. S., Gonçalves Ageitos, M., Ginoux, P., Balkanski, Y., Green, R. O., Kalashnikova, O., Kok, J. F., Obiso, V., Paynter, D., and Thompson, D. R.: Quantifying the range of the dust direct radiative effect due to source mineralogy uncertainty, Atmos. Chem. Phys., 21, 3973–4005, https://doi.org/10.5194/acp-21-3973-2021, 2021.

***Reply:*** *We have included some additional and alternative references according to the reviewer suggestions.*

Line 83: The authors may also want to explicitly identify which ESMs are still using OPAC data.
***Reply:*** *We have now included the ESMs that we are aware that still is using the OPAC data.*

Line 100: Lastly, clarification is needed regarding the size range of super-coarse particles. Kok et al. (2017) consider only particles smaller than 20 µm.

***Reply:*** *Thanks for pointing out the lack of clarity regarding the particle size definition. We now include how we define "super-coarse" and clarify that the Kok et al. (2017) study only consider particle size up to 20µm.*
However, later observations have shown that coarse to super-coarse dust(> 10 µm) sensu Adebiyi et al. (2023a) > 10, ≤ 62.5 µm is transported in non-negligible quantities further than expected (e.g., Ryder et al.,2018; Adebiyi et al., 2023a). . Including Kok et al. (2017) showed that including super-coarse particles in ESMs has been shown to reduce up to 20µm reduced the TOA DuERE by 50% (from −0.46 to −0.2 W m−2) due to the shift of the PSD to larger sizes, reducing SW extinction while increasing LW warming(Kok et al., 2017).